# Impact of international travel and diarrhea on gut microbiome and resistome dynamics

**Manish Boolchandani** [1,2], **Kevin S. Blake** [1,2], **Drake H. Tilley**[3],
**Miguel M. Cabada**[4,5], **Drew J. Schwartz** [1,6,7,8,9], **Sanket Patel**[1,2],
**Maria Luisa Morales**[5], **Rina Meza**[3], **Giselle Soto**[3], **Sandra D. Isidean**[10,11],
**Chad K. Porter**[10], **Mark P. Simons**[3,10] ✉ **& Gautam Dantas** [1,2,7,12] ✉

International travel contributes to the global spread of antimicrobial resistance. Travelers' diarrhea exacerbates the risk of acquiring multidrug-resistant organisms and can lead to persistent gastrointestinal disturbance post-travel. However, little is known about the impact of diarrhea on travelers' gut microbiomes, and the dynamics of these changes throughout travel. Here, we assembled a cohort of 159 international students visiting the Andean city of Cusco, Peru and applied next-generation sequencing techniques to 718 longitudinally-collected stool samples. We find that gut microbiome composition changed significantly throughout travel, but taxonomic diversity remained stable. However, diarrhea disrupted this stability and resulted in an increased abundance of antimicrobial resistance genes that can remain high for weeks. We also identified taxa differentially abundant between diarrheal and non-diarrheal samples, which were used to develop a classification model that distinguishes between these disease states. Additionally, we sequenced the genomes of 212 diarrheagenic *Escherichia coli* isolates and found those from travelers who experienced diarrhea encoded more antimicrobial resistance genes than those who did not. In this work, we find the gut microbiomes of international travelers' are resilient to dysbiosis; however, they are also susceptible to colonization by multidrug-resistant bacteria, a risk that is more pronounced in travelers with diarrhea.

The gut microbiota plays a critical role in deterring colonization by exogenous microorganisms[1]. International travel to destinations with high-infectious disease burdens can alter the gut microbiome and lead to the acquisition of multidrug-resistant organisms (MDRO)[2], which

may then spread locally when the traveler returns home[3,4]. Travelers' diarrhea affects 10–70% of travelers from low infectious disease-risk countries visiting middle- and high-risk destinations[5–8]. Diarrhea can disrupt travel plans, lead to long-term chronic health consequences

[1]The Edison Family Center for Genome Sciences and Systems Biology, Washington University School of Medicine, St. Louis, MO, USA. [2]Department of Pathology and Immunology, Washington University School of Medicine, St. Louis, MO, USA. [3]Naval Medical Research Unit No. 6, Callao, Lima, Peru. [4]Department of Internal Medicine, Division of Infectious Diseases, University of Texas Medical Branch, Galveston, TX, USA. [5]Cusco Branch – Tropical Medicine Institute, Universidad Peruana Cayetano Heredia, Lima, Peru. [6]Department of Pediatrics, Division of Infectious Diseases, Washington University School of Medicine, St. Louis, MO, USA. [7]Department of Molecular Microbiology, Washington University School of Medicine, St. Louis, MO, USA. [8]Department of Obstetrics and Gynecology, Washington University in St. Louis, St. Louis, MO, USA. [9]Center for Women's Infectious Diseases Research, Washington University in St. Louis, St. Louis, MO, USA. [10]Naval Medical Research Center, Silver Spring, MD, USA. [11]Henry M. Jackson Foundation for the Advancement of Military Medicine, Inc, Bethesda, MD, USA. [12]Department of Biomedical Engineering, Washington University in St. Louis, St. Louis, MO, USA. ✉e-mail: mark.p.simons.mil@health.mil; dantas@wustl.edu

such as post-infectious irritable bowel syndrome, and can increase risk for MDRO colonization[6,9]. As travelers' diarrhea is predominately caused by bacterial etiologies[6], antimicrobials are routinely used for treatment and prophylaxis[10]; however, such use compounds the risk for MDRO colonization. For example, Arcilla et al. reported 34% of Dutch travelers acquired extended-spectrum beta-lactamase (ESBL)-producing Enterobacteriaceae during travel with carriage lasting up to 12 months post-travel and observed transmission to non-traveling household members[4]. Importantly, those who developed diarrhea or took antibiotics during their stay abroad were more likely to acquire these ESBL-producing Enterobacteriaceae[4].

Previous studies on the impact of international travel on gut microbiome changes and MDRO acquisition have been largely limited to the collection of pre- and post-travel stool samples or have focused on select cultivable antimicrobial-resistant (AMR) organisms[2–4,11–14]. Thus, there is limited understanding about how the taxonomic and functional architecture of the gut microbiome changes during travel, how these are affected by diarrheal episodes, and how those dynamics influence MDRO acquisition and antimicrobial resistance gene (ARG) carriage. For example, following acute perturbation by infectious diarrhea, the gut microbiomes of non-traveling populations return to a healthy pre-diarrheal state within 1 month[15]; however, the gut microbiomes of international travelers are subject to additional chronic perturbations (e.g., different microbial ecology, dietary changes) which may shift the microbiome over time towards a more local state[16]. We hypothesize these chronic perturbations prevent the microbiome of international travelers from returning to a pre-diarrheal state, and that diarrhea markedly increases the rate of divergence from baseline. These dynamics can only be assessed with longitudinal studies which sample travelers' microbiome throughout the length of stay. In addition, while we know that international travel results in acquisition of select pathogenic MDROs[3], it is less clear how travel affects the overall carriage of ARGs in the broader microbiome. We hypothesize that length of stay in middle- and high-risk destinations are directly related to the prevalence of ARGs harbored by the gut microbiota, and—as with microbiome divergence—diarrhea accelerates this rate of increase. Analyzing the entire microbiome and resistome enables a comprehensive evaluation and stratification of travel risks, such as duration in-country, microbiome features, total ARG content, and occurrence of diarrhea. In this work, we address these questions by applying whole metagenome and whole genome sequencing to an extensive collection of longitudinally collected fecal samples and cultured bacteria from travelers from multiple countries visiting a common international destination. We show that the gut microbiomes of international travelers' are resilient to dysbiosis; however, they are susceptible to colonization by multidrug-resistant bacteria, a risk that is more pronounced in travelers with diarrhea.

## Results

### Overview of international travelers cohort

We assembled a prospective cohort of 159 international students (60% female, 40% male; median age: 24 years; age range: 18–65 years) attending the Amauta Spanish language school in the city of Cusco, Peru between June 2012 and July 2016. Participants originated from 16 low infectious disease-risk countries, and their median duration of participation was 35 days (interquartile range (IQR): 33 days; range: 2–173 days) (Fig. 1a). Stool samples were collected upon enrollment within 48 h of their arrival to Cusco and weekly thereafter, with additional samples collected during diarrheal events. The 113 individuals who experienced at least one episode of diarrhea during their stay (defined as one or more semi-liquid or watery bowel movements associated with the presence of gastrointestinal symptoms, see

Methods) were retrospectively classified into the "Travelers who experienced Diarrhea" (TD) group, while the 46 individuals who did not experience diarrhea were classified into the "Healthy Travelers" (HT) group (Fig. 1b). In total, we collected 718 stool samples, comprised of 144 diarrheal and 574 non-diarrheal sample types (HT $n = 212$, TD $n = 362$), with a median of four samples per individual (range: 1–22; Supplementary Fig. 1a–d). Additionally, we sub-divided TD subjects' samples into: non-diarrheal samples collected before the subject experienced diarrhea (PreTD), diarrheal samples (TD), and non-diarrheal samples taken after the subject experienced diarrhea (PostTD). All HT samples are non-diarrheal. Most individuals in the TD group experienced their first diarrheal episode within 1 month of arrival (78/113, 69%). Detailed demographic, medical history, and dietary data were collected from each participant (see Methods), and cohort characteristics with a summary of metadata features are listed in Supplementary Table 1. We observed no significant differences between HT and TD subjects with respect to age, sex, trip duration, or other demographic factors in univariable logistic regression analyses (Supplementary Tables 1 and 2; Supplementary Fig. 1b, c).

To characterize the gut microbial communities of travelers, we performed whole metagenome shotgun sequencing on the 718 stool samples and generated taxonomic composition (using MetaPhlAn2[17]), and antibiotic resistome (using ShortBRED[18]) profiles (Fig. 1b, Supplementary Fig. 2a, b). These profiles were supplemented with multiplex PCR on 696 stool samples and cultures to identify common diarrheagenic pathogens; whole genome sequencing of 212 diarrheagenic Escherichia coli (DEC) isolates to evaluate phylogenetic diversity and ARG content (Fig. 1c, Supplementary Fig. 2c, d); antibiotic susceptibility testing on 169 DEC isolates to determine phenotypic resistance (Fig. 1d, Supplementary Fig. 2e); and the construction of 21 functional metagenomic libraries[19–23] from 210 stool samples to characterize the antibiotic resistome in a sequence- and culture-unbiased manner (Supplementary Fig. 2f).

Among all the metadata variables, inter-individual variation accounted for the largest variation (44–52%) in taxonomic and resistome profiles (Supplementary Table 3, see Methods). For the other metadata variables, we observed relatively small (up to 4%) but significant variation associated with stool grade, sample type, country of residence, and duration of stay after correcting for multiple-hypothesis testing (all FDR $P < 0.05$; full results in Supplementary Table 3). Pathogen presence (identified by multiplex PCR) and subject age were also significantly associated with taxonomic but not with resistome profiles, while sample collection time was significantly associated with resistome but not with taxonomic profiles (Fig. 1e, Supplementary Table 3). Therefore, we conclude that these significant features (stool grade, sample type, country of residence, duration of stay, pathogen presence, age, and sample collection time) have systematic effects on the microbial community.

### Impact of international travel and diarrhea on the gut microbiota

Our longitudinally-collected metagenomic samples enabled us to evaluate the short- and long-term changes to travelers' gut microbiomes. Overall, the taxonomic diversity of the travelers' gut microbiota was temporally stable throughout the duration of their stay. As determined by linear mixed effect models (LMM; subject as random effect), the Shannon diversity of all HT and TD subjects' sample types were stable over time and did not significantly differ from each other (Fig. 2a, LMM, $P > 0.05$, Supplementary Table 4a); however, the richness of PostTD non-diarrheal samples was significantly lower compared to HT samples over time (LMM, $P < 0.001$, Supplementary Table 4a). In addition, when we compared TD individuals' diarrheal samples (TD) with matched non-diarrheal samples collected within 2 weeks before (PreTD) and after diarrhea (PostTD), we observed a significant decrease in both richness (GLMM, $P < 0.001$) and Shannon

diversity (Fig. 2b, LMM, $P < 0.05$) following diarrhea (Supplementary Table 4b).

While taxonomic diversity remained relatively constant, all participants' gut microbial communities underwent significant restructuring throughout the course of their travel abroad. First, we compared the beta-diversity of participants' consecutive stool samples. We found

that individuals in the TD group had greater apparent variation and less stable microbial architecture throughout the length of their stay than those in the HT group (Fig. 2c, LMM $P < 0.001$, Supplementary Table 5a). Importantly, we observed that individuals with greater taxonomic diversity at baseline were significantly more resilient to change than individuals with lower baseline diversity (LMM, $P$ (baseline

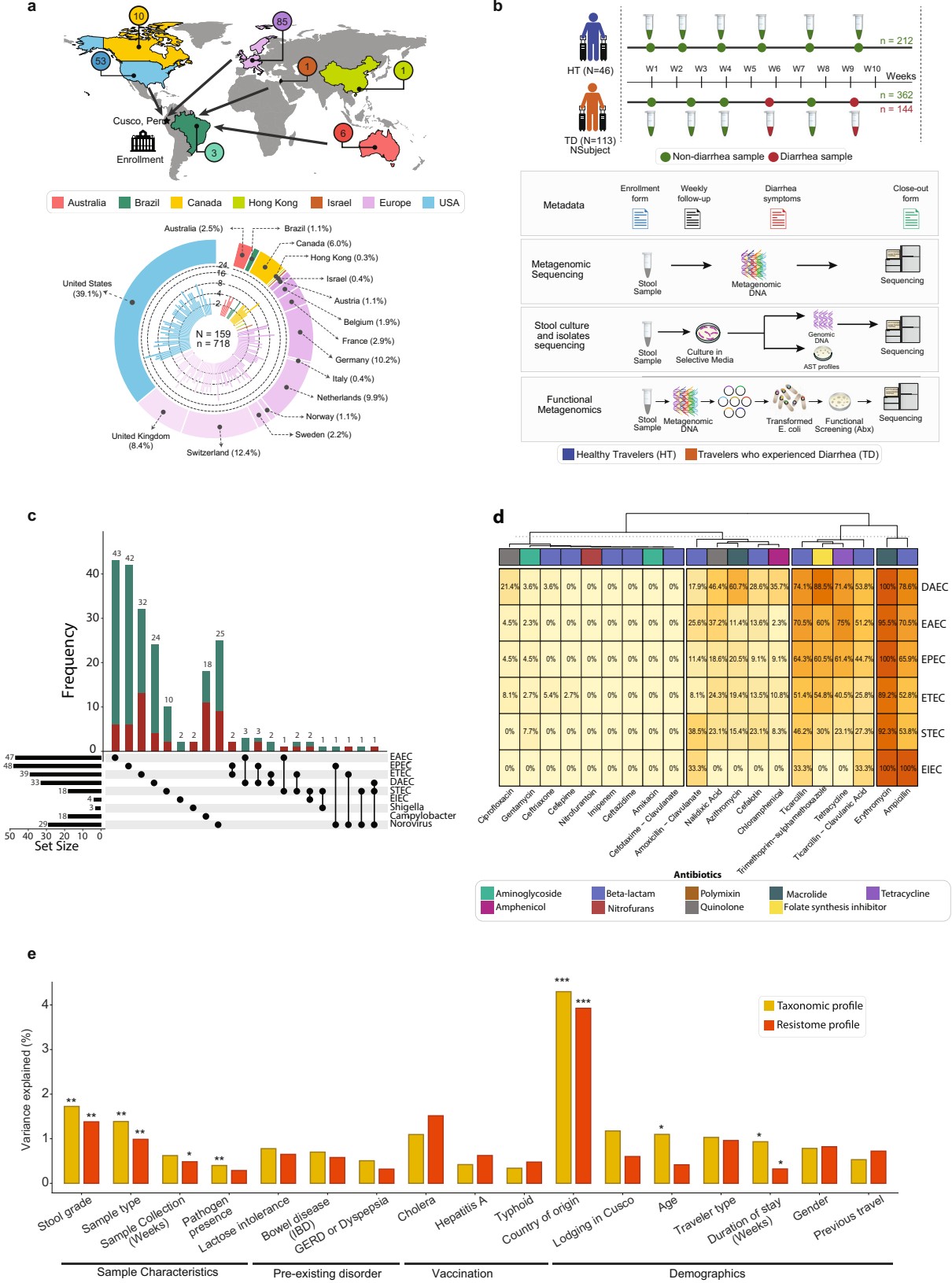

**Fig. 1 | Study design, data assembly, and meta-analysis for cohort of international travelers. a** Highlighted regions of the map show countries of residence for travelers in our cohort who visited the Andean city of Cusco, Peru between June 2012 and July 2016. Most traveled from North America or the European subcontinent. The Circos plot at the bottom shows the distribution of samples collected per individual, color-coded by their country of residence. **b** Sampling strategy and experimental design: Subjects who did not experience diarrhea during course of stay were classified as Healthy Travelers (HT, blue), while those who experienced at least one episode of diarrhea were classified as Travelers with Diarrhea (TD, orange). Collected stool samples were processed to obtain taxonomic, and resistome profiles using metagenomics sequencing. A subset of samples were cultured to obtain suspected diarrheagenic pathogens, tested for antibiotic susceptibility, and sequenced for detailed comparative genomic analysis.

Additionally, functional metagenomics libraries were prepared and screened against 17 antibiotics to characterize resistomes in the stool samples. **c** Upset plot depicting the frequency of diarrheagenic pathogens that were detected using multiplex PCR in non-diarrhea (green) and diarrhea samples (red). **d** Antibiotic susceptibility testing of diarrheagenic *E. coli* isolates against 20 antibiotics. Percentage and coloring of heatmap shows the proportion of isolates in a DEC pathotype resistant to the given antibiotic, with colored bar indicating antibiotic class. **d** PERMANOVA test quantifying the total variance explained by metadata variables in the taxonomic (MetaPhlAn2[17]) and resistome (ShortBRED[18]) profiles. The *y* axis indicates the total variance explained and the *x* axis represents different metadata variables that were tested (*$P < 0.05$; **$P < 0.01$; ***$P < 0.001$). Underlying data are provided in the Source Data file.

Shannon index) <0.001; Supplementary Table 5a). This is in-line with previous reports indicating that microbial diversity is an important contributor to overall intra-subject microbial stability and colonization resistance to enteric pathogens, with a lower pre-travel diversity being significantly associated with increased susceptibility to infection[24]. Next, to quantify divergence from the baseline gut microbial composition we compared the beta-diversity, measured by Bray–Curtis dissimilarity and the Jaccard index, of each travelers' samples to their first-week baseline sample. We found that the taxonomic composition of participants in both the HT and TD groups changed significantly over the length of stay (Fig. 2d, LMM, *P*(Time_diff_days) <0.001, Supplementary Table 5b), likely as a consequence of travelers' exposure to the different environmental microbial milieu of Peru compared to their home country. Notably, principal coordinate analysis (PCoA) of Bray–Curtis dissimilarity between TD samples which were temporally matched with before, during, and after diarrhea showed marked heterogeneity, suggesting that inter-individual variability among the samples exceeds the effect of diarrhea-induced changes (Supplementary Fig. 3a). Nevertheless, we observed a weak association between samples based on whether they were collected before, during, or after diarrhea (Supplementary Fig. 3a, PERMANOVA, $P = 0.009$, $R^2 = 0.10$).

To evaluate large-scale perturbations in HT and TD subjects' gut microbiota, we searched for "microbiome shift" events. These are defined as when the Bray–Curtis dissimilarity between an individual's consecutive samples (within a week) is greater than the dissimilarity between individuals (Supplementary Fig. 3b, see Methods)[25]. Using this approach, we identified 141 shift events (34.5%, 141/408). TD individuals had a significantly higher proportion of shift events than HT subjects (TD: 40.4%, 113/280; HT: 21.9%, 28/128; Fisher exact, $P < 0.001$; Supplementary Table 6), and the majority of TD individuals' shift events occurred during a diarrheal episode (67.3%, 76/113; Supplementary Table 6). These findings suggest that HT and TD subjects have similar gut microbial stability patterns, but large-scale disruptions occur most frequently during diarrhea.

While prior studies on infectious diarrhea among non-traveling native populations suggest an orderly reversal to the pre-diarrhea state within 1 month of the diarrheal episode[15], our cohort's microbial compositions increasingly diverge from baseline over the course of individuals' stays (Fig. 2d, LMM $P < 0.001$), and did not return to baseline at least for the duration of their stay. Thus, we sought to determine how experiencing diarrhea affects divergence from baseline, and whether rates of divergence might be a predictor of who will get diarrhea. We compared the Bray–Curtis dissimilarities of HT and TD subjects' 1st week baseline sample with a non-diarrheal sample collected 1 month later. We further subdivided the TD group between those who experienced diarrhea before 1 month (Early TD) or after 1 month (Late TD). Early TD subjects had significantly higher dissimilarity after 1 month of travel than the other groups (Fig. 2e, Wilcoxon signed-rank test, $P < 0.01$). This suggests that while all subjects' microbiomes continuously diverge from baseline during travel,

diarrhea is an impactful perturbation that significantly increases this divergence. Further, we observed no significant difference between HT and Late TD subjects (Fig. 2e, Wilcoxon signed-rank test $P > 0.05$). This suggests that the divergence of Late TD subjects prior to diarrhea is indistinguishable from those who will not get diarrhea (HT), thereby precluding the use of diversity metrics alone as an early predictor of who will get diarrhea. This motivated us to quantify the taxonomic differences between subject groups at higher resolutions.

We found that diarrhea significantly altered the composition of major gut microbial phyla (Supplementary Fig. 3c). Diarrheal samples were characterized by enrichment for Proteobacteria and Bacteroidetes, and a depletion of Firmicutes, resulting in a lower Firmicutes-to-Bacteroidetes ratio (Supplementary Fig. 3d, Wilcoxon signed-rank test, $P < 0.001$). This is consistent with other gastrointestinal diseases and is associated with a dysbiotic gut microbial architecture[26]. However, we observed no significant difference in the Firmicutes-to-Bacteroidetes ratio between the before (PreTD) and after diarrhea (PostTD) samples (Supplementary Fig. 3d, Wilcoxon signed-rank test, $P > 0.05$), suggesting that, while the overall composition does not return to the pre-diarrheal state, these specific phyla quickly recover. At species-level resolution, we identified 39 differentially abundant species associated with diarrheal and non-diarrheal samples using multivariable regression models (using MaAsLin2[27]) (Fig. 2f–g, Supplementary Fig. 4). In diarrheal samples, we found an elevated relative abundance of bacteria with known diarrhea-causing pathogenic strains (e.g., *E. coli*, and *Shigella* spp.). Additionally, we saw an increased abundance of Proteobacteria (e.g., *Bilophila* spp., *Sutterella wadsworthensis*, *Parasutterella excrementihominis*, and members of the *Burkholderiales* order (Fig. 2f–g). Previous studies have linked these taxa to host physiology and health outcomes[28–32], such as an increased abundance of *Bilophila* spp. being associated with inflammatory bowel disease[29], and increased abundance of *Sutterella wadsworthensis* being associated with ulcerative colitis etiology and fecal microbiota transfer treatment failure[30–32]. We also observed a significant increase in the relative abundance of several taxa within the Bacteroidetes phylum (e.g., *Odoribacter splanchnicus, Bacteroides fragilis, Bacteroides vulgatus*), consistent with previous reports on diarrheal diseases[33,34]. Bacteroides are major producers of sphingolipids that regulate inflammation and immunity in the human gut[35], and a bloom during diarrhea may reflect an effort to restore microbial homeostasis[36,37]. In contrast, several taxa belonging to the Firmicutes phylum were depleted in diarrheal samples (e.g., *Ruminococcus bromii, Eubacterium rectale, Clostridium bartlettii, Coprococcus* spp.). Many Firmicutes are residents of healthy guts and are known to play key functions, including maintenance of gut barrier function and digestion of complex polysaccharides by producing short-chain fatty acids[38]. A decreased representation of Firmicutes has also been associated with other gastrointestinal diseases, including ulcerative colitis and Crohn's disease[39].

In concordance with prior studies[40–42], no identifiable etiologic agent was detected in ~50% of the diarrheal samples in our study, even

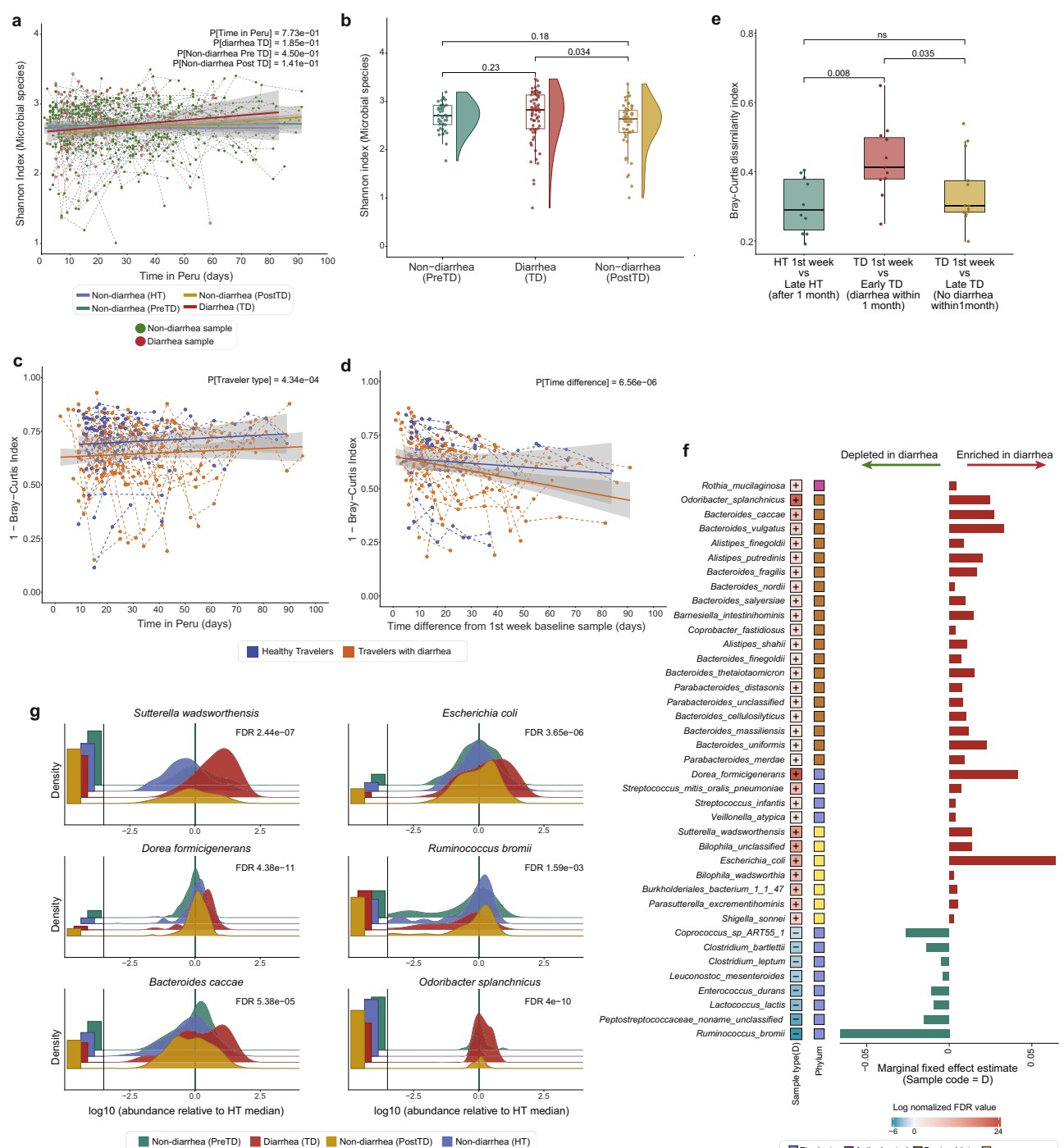

**Fig. 2 | Travelers' gut microbiome compositions change throughout travel, with diarrhea-causing large-scale disruptions. a** Taxonomic diversity (Shannon index) of subjects' microbiomes throughout their length of stay, as days post-arrival. Points are individual samples, colored by sample type with dotted lines connecting samples from the same subject. Solid lines show the best fit line for different sample types: HT (blue), diarrheal (red), non-diarrheal preTD (green), and postTD samples (yellow) (*n* = 617, LMM, all *P* > 0.05), and the gray shading represents 95% confidence interval (CI). **b** Boxplots of the taxonomic diversity of subject-matched diarrheal and non-diarrheal samples collected within 2 weeks (*n* = 171, LMM, P[DiarrheaTD vs Non-diarrhea PostTD] = 3.4e-02). **c** Bray–Curtis dissimilarities between consecutive samples from each subject, plotted throughout their length of stay with dotted lines connecting samples from the same subject. Solid lines show the best fit for different traveler types (*n* = 291, LMM, P[Traveler type] <0.001) **d** Bray–Curtis dissimilarities between each subjects' samples and their 1st-week baseline sample (*n* = 291, LMM, P[Time difference] <0.001) **e** Boxplots of Bray–Curtis dissimilarities calculated between 1st week baseline sample and a

non-diarrheal sample collected 1 month after arrival in HT and TD subjects'. TD subjects were further sub-divided into Early TD (who experienced diarrhea <1 month of arrival) and Late TD (who experienced diarrhea >1 month after arrival) (*n* = 36, Wilcoxon test). Boxes in the boxplots show median and quartiles; error bars extend to the values within 1.5 interquartile range. **f** Microbial species significantly associated with diarrhea samples detected using MaAsLin2[27] where subjects were included as random effects and other metadata variables as fixed effects. The significant associations were corrected for multiple-hypothesis testing using Benjamini–Hochberg method with FDR < 0.25. The first column depicts the log normalized FDR value calculated by −*sign(coeff)\*log(qval)*, the second column shows the phylum and the barplot shows the effect size of each species. **g** The density ridgeline plot of significantly associated species in different sample types normalized by the median relative abundance of non-diarrheal HT samples. Left barplot, fraction of samples below detection limit. Underlying data are provided in the Source Data file.

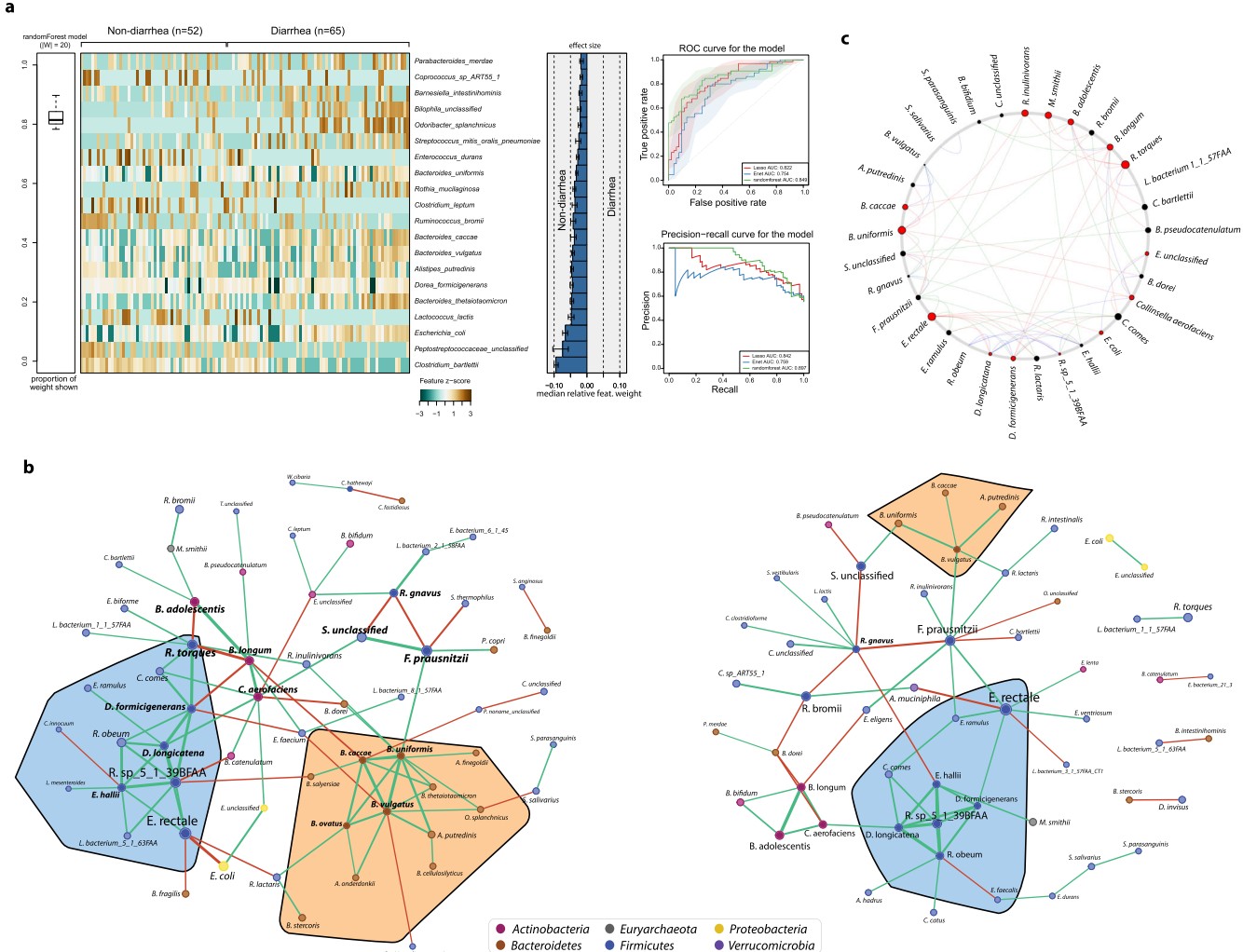

**Fig. 3 | Development of machine learning classifier that uses species abundances to distinguish between diarrheal and non-diarrheal sample types. a** The heatmap showing the relative abundances of top 20 discriminatory taxa between diarrheal and non-diarrheal PreTD types. Barplot depicts the effect size of the discriminatory taxa obtained from Random forest model. Cross-validation accuracy depicted as AUC-ROC and Precision-recall curve for the models built with three independent methods: ElasticNet (blue), Lasso (red) and Random Forest (green). **b** Co-occurrence network of diarrheal (left) and non-diarrheal (right) samples. The sizes of the nodes are proportional to the mean relative abundances of the species. The edges between two nodes represent significant correlations (green: positive; red: negative) between the two species. The orange highlighted region shows Bacteroidetes that are enriched in diarrhea (left) compared to non-diarrhea (right) network. The blue-highlighted region shows a conserved relationship among Firmicutes in both networks. **c** Circos plot depicting the common subnetwork species between diarrheal and healthy sample networks. Node sizes are proportional to their scaled NESH score obtained from Netshift[45], and species that gained connectivity in the diarrhea network compared to the healthy network are colored red. Connectivity changes between the common subnetworks are represented by connections between these species (red: connections that are unique to diarrhea samples network; blue: connections that are shared in both networks; green: connections that are unique to non-diarrheal samples network). Underlying data are provided in the Source Data file.

when using multiplex PCR. Although we observed significant alterations for a number of species between the diarrheal and non-diarrheal samples (Fig. 2f), these changes could be either a signature of dysbiotic gut or disease-specific alteration. The disease-specific microbiome changes may reflect potential pathogens or pathobionts exploiting diarrhea-induced changes in the intestinal milieu. Thus, to discern diarrhea-specific microbial signatures, we subsampled the diarrheal samples (TD) and non-diarrheal samples collected before diarrhea episode (PreTD) and employed machine learning methods on the abundances of key taxa. Specifically, we applied three approaches viz. ElasticNet, Lasso, and Random Forest (RF) using SIAMCAT[43]. Among these, the RF model had the best accuracy (84.9%) and precision-recall (89.7%) (Fig. 3a). Consistent with prior observations, 8 of these 20 top discriminatory species belonged to the Bacteroides family. The top five species with the largest effect sizes were *C. bartlettii*, *E. coli*, *Peptostreptococcaceae unclassified*, *L. lactis*, and *B. thetaiotamicron*. Apart

from *E. coli*, the other four species are key commensal species that were depleted during diarrhea.

Lastly, to understand how these individual taxa interact with each other during diarrhea, we built two unsupervised co-occurrence networks from diarrheal and non-diarrheal samples (using SparCC[44]) (Fig. 3b) and then compared these correlation networks using Netshift[45] (Fig. 3c) to identify "driver" taxa responsible for differences. This identified 14 taxa that showed significant shifts in their interactions in diarrhea compared to non-diarrheal samples (Fig. 3c, highlighted in red). Among them, *E. coli* and *Escherichia* spp. *Unclassified* were enriched during diarrhea, gained connectivity, and were negatively associated with key commensal species such as *Eubacterium rectale*. Bacteroidetes species such as *B. uniformis* and *B. caccae* were also enriched, gained interactions, and formed a close sub-network of positive interactions with other *Bacteroides* species (Fig. 3b, highlighted in orange). During diarrhea, other commensal microbes

including *F. prausnitzii* and *Ruminococcus bromii* were depleted and lost their connectivity to other microbial species (Supplementary Table 7). While the gut microbiome underwent major remodeling during diarrhea, a sub-network built from *D. longicatena*, *D. formicigenerans*, *E. hallii*, *R. obeum*, and *R. sp_5_1_39BFAA* remained conserved with no change in interactions between each other (Fig. 3b, highlighted in blue). Notably, many of these interconnected and differentially interacting taxa were also found to be differentially abundant by the LME (MaAsLin2[27]) (Fig. 2f), as well discriminatory between PreTD and TD sample types by the RF model (SIAMCAT[43]) (Fig. 3a).

### Temporal dynamics of antibiotic resistance gene diversity and abundance

Having characterized the dynamics of changes to the gut microbiome during travel, we next sought to determine if similar trends characterize the travelers' gut antibiotic resistomes. To capture both known and potentially sequence-novel functional ARGs encoded by international travelers' gut microbiota in a high-throughput, sequence- and culture-unbiased manner, we constructed 21 functional metagenomic libraries (representing 89.5 GB) from 210 stool samples randomly selected from the full set of 718 samples (see Methods). These libraries were screened against 17 antibiotics (Supplementary Data 1), with resistance found against all antibiotics screened except for ciprofloxacin and meropenem (Supplementary Fig. 2f). Resistance was most abundant for trimethoprim and tetracyclines, and lowest for colistin and 3rd/4th generation cephalosporins. The metagenomic inserts from these resistant transformants were sequenced, assembled, and annotated for AR function using our previously published pipeline[18,19] (see Methods). This yielded 2,065 unique ARG sequences, expanding the catalog of known ARGs harbored by the gut microbiota. We then built an extensive ShortBRED[18] AR protein sequence marker database by incorporating ARG sequences from this cohort ($n = 2065$), 12 other published functional metagenomic studies (Supplementary Table 8), and two curated ARG databases (CARD[46] v2.2.0; NCBI-AMR[47] v1.0), resulting in a database consisting of 6,594 unique marker sequences representing 2,314 ARG families (Supplementary Data 2). The relative abundance of ARGs in the sequenced metagenome of each stool sample was then quantified by mapping the shotgun data to the ShortBRED marker database.

The diversity of gut resistomes harbored by travelers' gut microbiota demonstrated temporal stability for the duration of travel. Similar to microbiome diversity, Shannon diversity of ARGs from HT and TD subjects' samples showed no significant change over the length of travel (Fig. 4a; LMM, $P > 0.05$, Supplementary Table 9a). However, diarrheal samples ARG richness significantly increased over time compared to HT samples (LMM, $P < 0.01$; Supplementary Table 9a), suggesting subjects who experienced diarrhea later had a greater diversity of ARGs. We also did not observe a significant change in cumulative ARG abundance (RPKM) over length of travel (Fig. 4b, LMM, $P > 0.05$, Supplementary Table 9a). We observed a marked increase in ARG richness (LMM, $P < 0.001$) and abundance (LMM, $P < 0.001$) among individuals who reported using antibiotics in the preceding week (Supplementary Table 9b). When comparing the beta-diversity of individuals' gut resistomes from consecutive samples (by Jaccard and Bray–Curtis index), we observed TD subjects' samples were significantly more dissimilar than HT, indicating the TD resistome is less stable (Supplementary Fig. 5a, LMM, $P < 0.05$; Supplementary Table 10a). We also observed significantly increased similarity by Jaccard index in later samples, suggesting greater resistome restructuring occurs earlier in travel (LMM $P < 0.05$; Supplementary Table 10a). Further, when comparing subjects' samples to their 1st week baseline sample, we observed significantly increased dissimilarity with increasing time differences between the samples compared, indicating restructuring occurs throughout the length of travel (Supplementary Fig. 5b, LMM, $P < 0.05$; Supplementary Table 10b).

However, diarrhea transiently increases ARG diversity and abundance. Comparison of diarrheal samples with matched non-diarrheal samples collected within 2 weeks (PreTD) showed diarrheal samples had significantly increased ARG richness (GLMM, $P$ (DiarrheaTD) <0.001), Shannon diversity (Fig. 4c, LMM, P (DiarrheaTD) <0.001), and cumulative ARG abundance (Fig. 4d, LMM, $P$ (DiarrheaTD) <0.001, Supplementary Table 9c). While ARG richness and diversity recovered to PreTD levels within 2 weeks after diarrhea (PostTD), ARG abundance remained significantly high (Fig. 4d; LMM, $P < 0.001$, Supplementary Table 9c). Further, we find this increase in abundance of ARGs during diarrhea was significantly correlated with a decrease in microbial diversity at the species-level (Fig. 4e, LMM, P < 0.001) but an increase in relative abundance of *Enterobacteriaceae* (Fig. 4e, LMM, $P < 0.01$). This suggests that the majority of AMR determinants in diarrheal samples are likely concentrated within Enterobacteriaceae species. Additionally, we observed a significant negative correlation of ARG abundance with species belonging to *Ruminococcaceae* (LMM, $P < 0.001$), *Eubacteriaceae* (LMM, $P < 0.001$), *Coriobacteriaceae* (LMM, $P < 0.01$), and *Bifidobacteriaceae* (LMM, $P < 0.01$) (Supplementary Fig. 5c). Finally, we sought to identify specific ARGs that were differentially abundant in diarrheal samples using MaAsLin2[27]. Diarrheal samples had an increased abundance of antibiotic efflux pumps, β-lactamases (Class A and Class C), and aminoglycoside resistance genes, while non-diarrheal samples were enriched for tetracycline ribosomal protection genes (Fig. 4f–g, Supplementary Fig. 5d, Supplementary Fig. 6). Importantly, many of these ARG families that are enriched in diarrhea samples are commonly encoded in Enterobacteriaceae species.

### Phenotypic and phylogenetic analyses of diarrheagenic pathogens

As bacterial etiologies are the predominant cause of travelers' diarrhea[48], we evaluated both diarrheal and non-diarrheal stool samples for the presence of common diarrheagenic pathogens. We performed multiplex PCR on 696 stool samples and cultures to detect *Campylobacter* spp., *Shigella* spp., norovirus (GI, GII), and six strains of diarrheagenic *E. coli* (DEC) (Supplementary Table 11; Supplementary Information). At least one diarrheagenic pathogen was detected in 217/696 samples (31.2% overall; 62/142, 43.7% diarrheal; 155/554, 28.0% non-diarrheal). No pathogen was detected in 80/144 (55.6%) diarrheal samples, consistent with the previous reports[7]. DEC were the most common group of pathogens detected (193/217, 88.9%), followed by *Norovirus* (29/217, 13.4%), *Campylobacter* spp. (18/217, 8.3%), and *Shigella* spp. (3/217, 1.4%) (Fig. 1c; Supplementary Table 11; Supplementary Information). Simultaneous detection of ≥2 enteropathogens was observed (24/217, 11.1%), predominantly for different DEC strain types (19/217, 8.8%). To analyze DECs directly, we cultured 212 DEC isolates from 195 fecal samples (157 isolates from 149 non-diarrheal samples, and 55 isolates from 46 diarrheal samples). To assess AMR phenotypes, we performed antimicrobial susceptibility testing on 169 DEC isolates against 20 antibiotics belonging to 12 antimicrobial classes (Supplementary Table 12). The majority of DEC isolates (66.9%, $n = 113$) were multidrug-resistant (MDR; defined as resistance to ≥1 antimicrobial agent in ≥3 antimicrobial classes; Supplementary Fig. 2e). We observed moderate rates of resistance to azithromycin (AZM; 23.7%, $n = 40$) and ciprofloxacin (CIP; 7.7%, $n = 20$), two antibiotics recommended for the treatment of acute diarrhea[48] (Supplementary Fig. 2e).

Next, we sequenced the genomes of all 212 DEC isolates to investigate the genomic diversity, ARG content, and virulence potential of these enteropathogens. The draft assemblies were quality filtered and 23 isolates with poor assembly metrics were excluded (Supplementary Fig. 2c, d, see Methods). We analyzed the population structure of the quality filtered genomes, along with 40 publicly available *E. coli* genomes of diverse pathotypes, by constructing a core-genome (2216 genes, ≥95% identity) maximum-likelihood phylogenetic tree (using Roary[49] and RAxML[50]) (Fig. 5). The phylogenetic

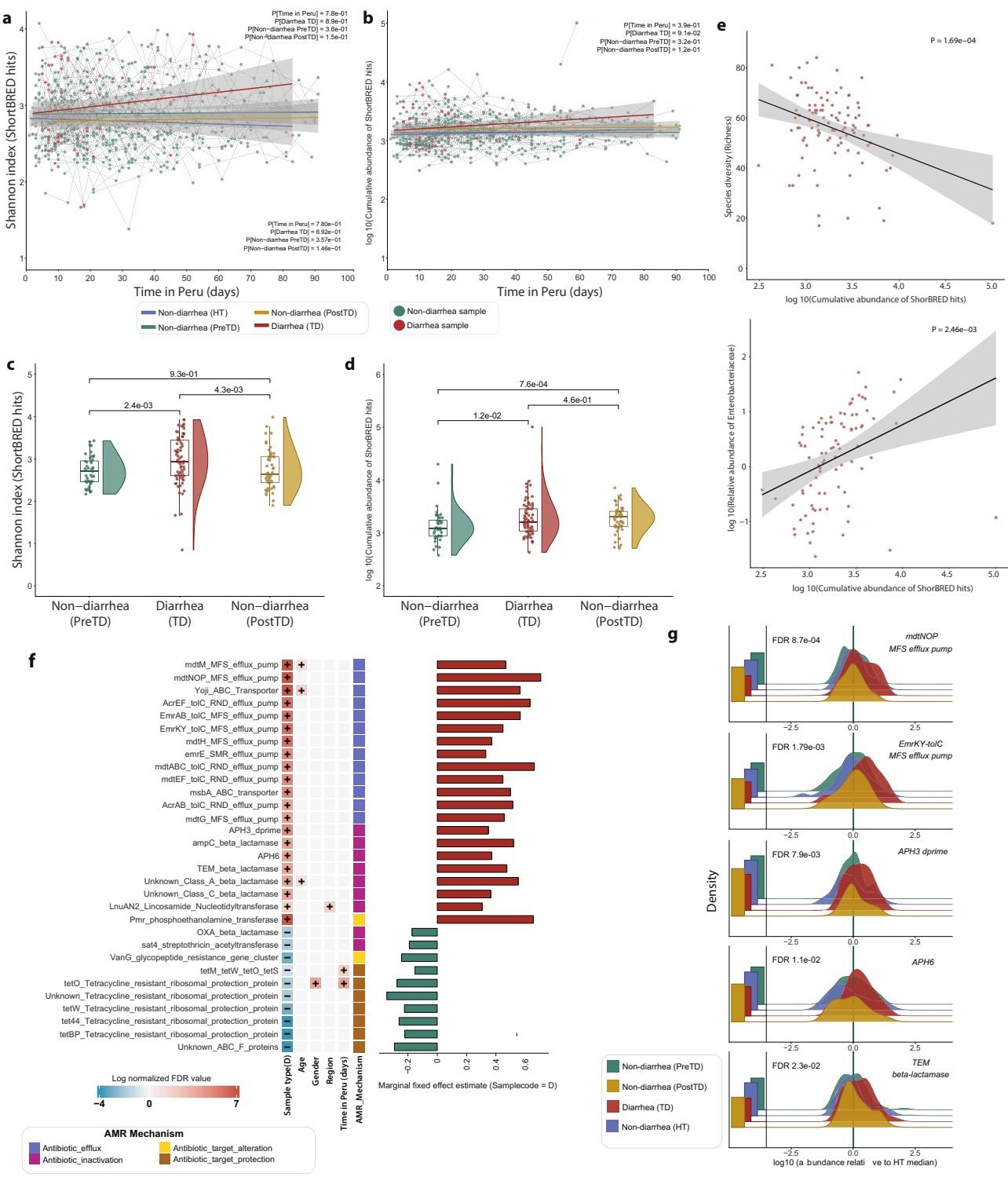

clusters were independently confirmed using BAPS[51]. Our DEC isolates grouped into 6 distinct clusters corresponding to phylogroups A, B1, B2, D, E, and F. The majority belonged to Clade A (106/189; 56.1%) and Clade B1 (56/189; 29.6%). No significant association between phylogroups and sample type (diarrheal vs. non-diarrheal) was observed (Fishers exact test, $P > 0.05$), except for isolates belonging to Clade B2 and Clade F that were isolated only from non-diarrheal samples.

DEC strains isolated from non-diarrheal samples in TD subjects encoded more ARGs than isolates from HT subjects (Fig. 6a; Supplementary Fig. 7a, b) (non-diarrheal HT vs. non-diarrheal TD, Wilcox test,

$P < 0.05$). These isolates were significantly enriched for $bla_{\text{TEM-148}}$, $sul2$, $aph6$, and $tetA$ genes (Fig. 6a). Notably, several DEC strains isolated from TD subjects also carried other clinically significant AMR genes including, ESBL genes ($bla_{\text{CTX-M}}$, $n = 7$; $bla_{\text{OXA}}$, $n = 3$), AmpC ($bla_{\text{CMY}}$, $n = 1$), and $mcr-1$ ($n = 1$). Notably, mcr-1 was first described in the literature in 2016[52,53], whereas this $mcr-1$ harboring isolate was cultured in 2013. Colonization with multidrug-resistant DEC strains could have long-term consequences both during and after travel, such as infection by the pathobiont or horizontal transfer of its MDR genes to another pathogen leading to treatment failure or other adverse clinical

**Fig. 4 | Diarrhea increases ARG diversity and abundance. a** The alpha diversity (Shannon index) of subjects' resistomes throughout the length of their time in Peru, as days post-arrival. Points are individual fecal samples, colored by sample type (i.e., diarrhea or non-diarrhea), with dotted lines connecting samples from the same subject. Solid lines show the best fit for different traveler types: HT (blue), diarrheal (red), non-diarrheal pre-TD (green), and post-TD samples (yellow) ($n = 617$, LMM, $P > 0.05$) and the gray shading represents 95% confidence interval (CI). **b** Cumulative abundance of ARGs ($\log_{10}$ scale) over time. Points are individual samples with dotted lines connecting samples from the same subject. Solid lines show best fit with 95% CI (gray shading) of samples from different traveler types ($n = 617$, LMM, $P > 0.05$). **c** Boxplots of ARG diversity (Shannon index) of subject-matched diarrheal samples with non-diarrheal samples collected within 2 weeks of diarrhea. Accompanying violin plots show the distribution ($n = 171$ samples, two-sided Wilcoxon test). **d** Cumulative abundance of ARGs of subject-matched diarrheal samples with non-diarrheal samples collected within 2 weeks of diarrhea.

Accompanying violin plots show the distribution ($n = 171$ samples; two-sided Wilcoxon test). **e** Correlations between cumulative abundance of ARGs ($\log_{10}$ scale) and microbial species diversity (top) and relative abundance of *Enterobacteriaceae* ($\log_{10}$ scale) (LMM; all $P < 0.001$). **f** ARGs that enriched or depleted in diarrheal samples compared to non-diarrheal samples. The significant associations were detected by MaAsLin2[27] where other metadata variables (age, sex, sample type, region, length of stay, and antibiotics usage) were used as fixed effects in the LMM. **g** Relative abundance distribution of differentially abundant ARGs in different sample types normalized by the median relative abundance of non-diarrheal HT samples (LMM; all $P < 0.001$). Left barplot, fraction of samples below detection limit. Boxes in the boxplots show median and quartiles; error bars extend to the values within 1.5 interquartile range. $P$ values are multiple-hypothesis test corrected using Benjamini–Hochberg (FDR) method. Underlying data are provided in the Source Data file.

outcomes, and spread to new hosts during and after travel (e.g., household members, healthcare facilities)[3,4].

We also observed several ARG clusters associated with plasmids. Pairwise co-occurrence comparisons of accessory AR determinants in the same contig (within 5 kb) revealed highly interconnected groups of ARGs among DEC isolates (Fig. 6b). The most common cluster of ARGs frequently detected together ($bla_{TEM-148}$, $sul2$, $aph6$, $aph3$, $emrE$, and $tetA$) was also found to be significantly associated with plasmids (49%; 25/51; Fisher's exact test; FDR $P < 0.05$). Several other ARGs were also found in plasmid sequences but could not be evaluated because of their overall low frequency in the dataset (Supplementary Table 13). We next analyzed the genomic context of these ARGs to assess transmission risk. We identified mobile genetic elements (MGE) within 5 kb of the ARGs using MGEfinder[54] (Fig. 6c, Supplementary Fig. 8), suggesting these ARGs are able to transfer horizontally. We then built a co-occurrence network of the observed ARGs with MGEs and identified clusters that were more frequently observed together in the isolates. We identified four major clusters that include ARGs conferring resistance via different mechanisms (Fig. 6b). The genes $sul2$, $aph6$, $aph3$, and $bla_{TEM-148}$ were the most common group observed, occurring in 60 isolates (39.1%). These genes were often observed with the IS26 insertion element ($n = 12$) and Tn2 unit transposon ($n = 12$), as well as with dfrA8/dfrA14 genes ($n = 20$) and other mobile elements like IS903. Co-occurrence and co-transfer of these ARGs raise concerns over their potential expansion and dissemination.

We identified instances of temporary (1 timepoint) and persistent (≥2 timepoints) colonization by DEC isolates, as well as co-colonization of multiple DEC strain types in a single timepoint. For individuals with ≥2 DEC isolates collected from their longitudinal stool samples, we compared the phylogenetic relatedness of those isolates using StrainSifter[55] (Supplementary Fig. 9). The single-nucleotide variation (SNV) per megabase of isolates from the same individual was highly variable, ranging from 0 to 16,833 SNVs. For example, DEC isolates from HT-P041 on days 33, 39, and 45 had zero SNVs, suggesting a single strain of *E. coli* had persisted in that individual's gut throughout the two-week period (Supplementary Fig. 9; font colored yellow). In contrast, isolates cultured from the diarrheal sample of TD-P004 collected on day 77 differed by 9,187 SNVs per megabase, suggesting simultaneous colonization by distinct DEC strain types (Supplementary Fig. 8; font colored purple). Supporting the case for transient co-colonization, these isolates also showed distinct ARG and VF profiles. Coexistence of multiple DEC strains with distinct resistome profiles in a diarrheal sample can potentially affect the ability to treat diarrhea with antibiotics. Lastly, we calculated the pairwise SNV count from isolates collected longitudinally from the same subjects, and observed isolates from HT subjects had greater stability than those from TD subjects, measured as fewer SNVs (HT median pairwise SNV count = 5218; TD median SNV count = 8922; $t$ test $P < 0.05$). We speculate that, as with the gut microbiome, TD subjects' *E. coli* are similarly stable to HT

subjects except during diarrheal episodes when large-scale disruptions occur (as measured by shift events; Supplementary Table 6) and commensal *E. coli* are replaced by DEC. However, in this study, we do not have the high-resolution sampling around diarrheal episodes necessary to test this. We also note that these measurements of strain diversity, as well as counts of temporary and persistent colonization are likely an underestimate, as higher-resolution studies with greater sampling frequency have shown that transient colonization of international travelers can last just one or a few days[56], and strains could have persisted under the detection threshold of our culture methods.

## Discussion

Here we report a large longitudinal interrogation of host microbiota-pathogen interactions and acquisition of MDRO during international travel to a region with a high-infectious burden. We observed travelers' gut microbiome compositions diverging significantly from their baseline microbial architecture throughout the length of their stay, with the greatest change occurring during the first month of stay. This is likely explained by sustained exposure to a non-native environment with different microbial ecologies, as well as changes in diet, surrounding climate, and other environmental factors (e.g., altitude)[8,57–59]. Presumably, travelers' microbiomes were shifting towards a more local state wherein this divergence would stabilize; however, we cannot definitively claim this as our cohort does not include samples from the local population, nor was their stay as long enough as observed in other traveling populations which took 6–9 months to acclimate[16]. The microbiome diversity of HT subjects showed remarkable temporal stability, while those of TD subjects were less stable and had greater variation. The overall temporal stability of travelers' gut microbiomes was attributed to higher baseline microbial diversity. During diarrheal episodes we observed a dysbiotic microbial architecture marked by the enrichment of Bacteroidetes and Proteobacteria and contrasting depletion of Firmicutes, similar to the previous reports[26]. Unlike studies in native populations[15], the microbiomes of our travelers did not return to a pre-diarrheal state even after 1 month of the diarrheal episode, which is likely attributed to the microbiome's continual divergence due to the non-native environment. We also identified several taxa that were differentially abundant between non-diarrheal and diarrheal samples, which were then used to develop a classification model that can distinguish diarrheal from non-diarrheal disease states with high accuracy and precision-recall. This underscores the ability of microbiome-based biomarkers to differentiate between disease and healthy states; however, more studies using traveler cohorts visiting different geographical locations are needed to determine the generalizability of these features, and more mechanistic studies with model systems (e.g., microbiota-humanized gnotobiotic animals)[60–62] would be needed to elucidate the putative causal role of these species.

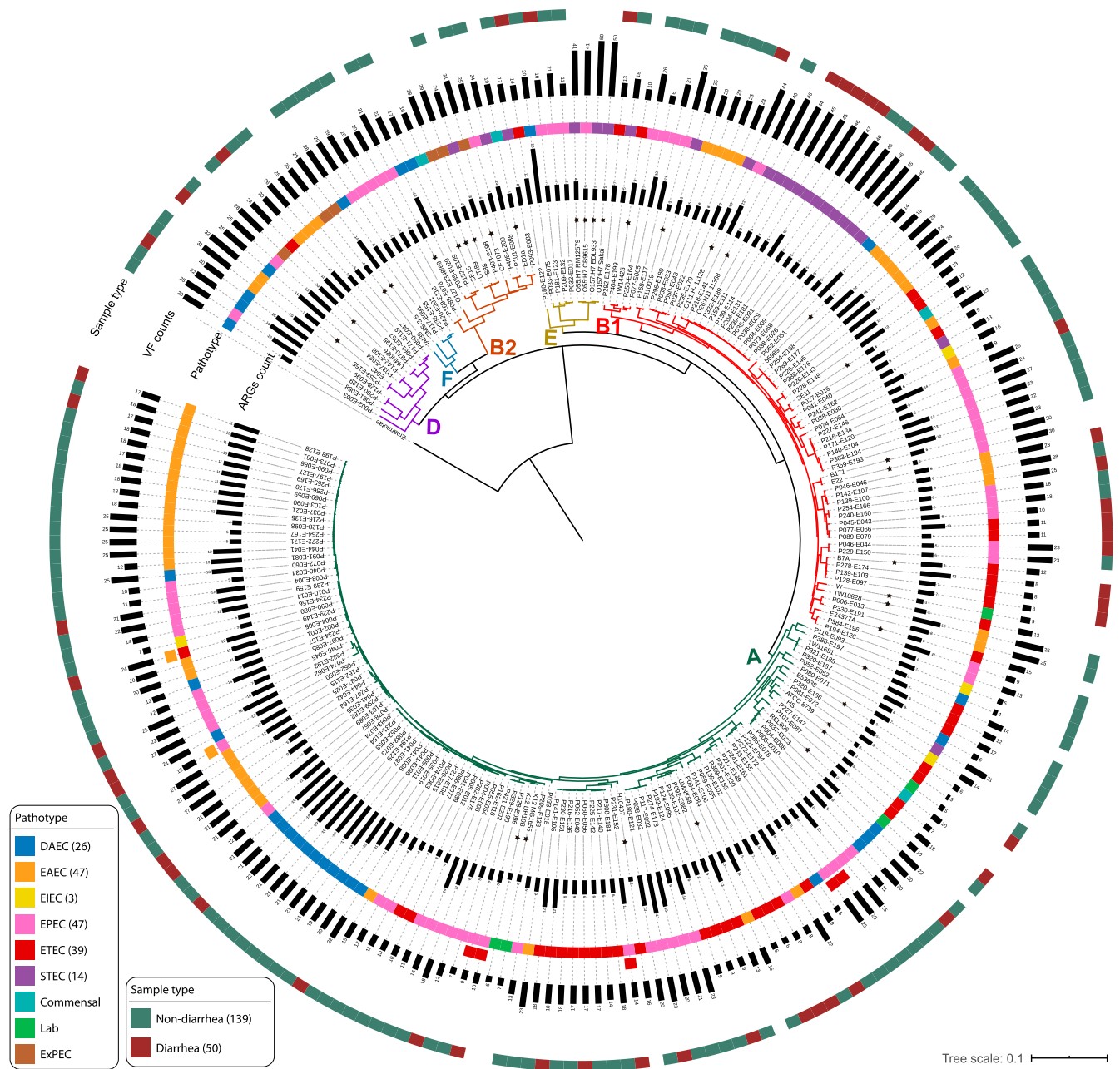

**Fig. 5 | Diarrheagenic *E. coli* isolates are phylogenetically diverse and encode many ARGs and VFs.** Phylogenetic tree inferred from core-genome alignment of 189 *E. coli* isolates (this study) and 40 published *E. coli* reference genomes (marked by *). Phylogroups are depicted as colored branches of the tree. Annotations, from inner to outer ring, denote: Gray barplots denote ARG count; *E. coli* pathotype (assigned by the presence/absence of specific VFs; Methods); VF count; and sample type (diarrheal or non-diarrheal). Underlying data are provided in the Source Data file.

By complementing metagenomic sequencing with functional metagenomics[20,63–65], we comprehensively assessed the ARGs harbored by travelers' gut microbiota. In contrast to other studies[2], we found that travelers' resistomes were temporally stable throughout their stay, with marginal increases noted in ARG abundance over time. This difference can be partially attributed to different travel destinations, as the country and region visited can have profound effects on ARG acquisition[3,66,67]. Nonetheless, we found that diarrheal events significantly altered travelers' microbiomes and resistomes, resulting in the enrichment of ARGs—particularly those encoded by Enterobacteriaceae species. This observation was corroborated by quantitative analysis of DEC (a key member of the Enterobacteriaceae family) isolates, where we found isolates from TD subjects encoded more resistance genes than those from HT subjects even during

asymptomatic periods. We further identified ARGs strongly associated with TD subjects and found a small cluster of plasmid-borne, resistance-conferring genes (*bla*$_{TEM-148}$, *sul2*, *aph6*, and *tetA*) likely able to be mobilized through horizontal gene transfer (HGT). Collectively, these observations suggest that travelers' diarrhea is a significant risk factor for increased carriage of MDR *E. coli* and ARGs in the microbiome. The acquisition of these MDROs can have long-term consequences for the traveler, such as HGT of ARGs to another pathogen that could potentially cause treatment failures, longer hospitalization stays, and additional impacts on public health systems as they spread to new hosts upon return from travel[68].

Although we identified antibiotic use for prophylaxis or treatment during travel as a significant risk factor for increased ARG carriage, we were unable to separate the effect of specific antibiotics. Controlled

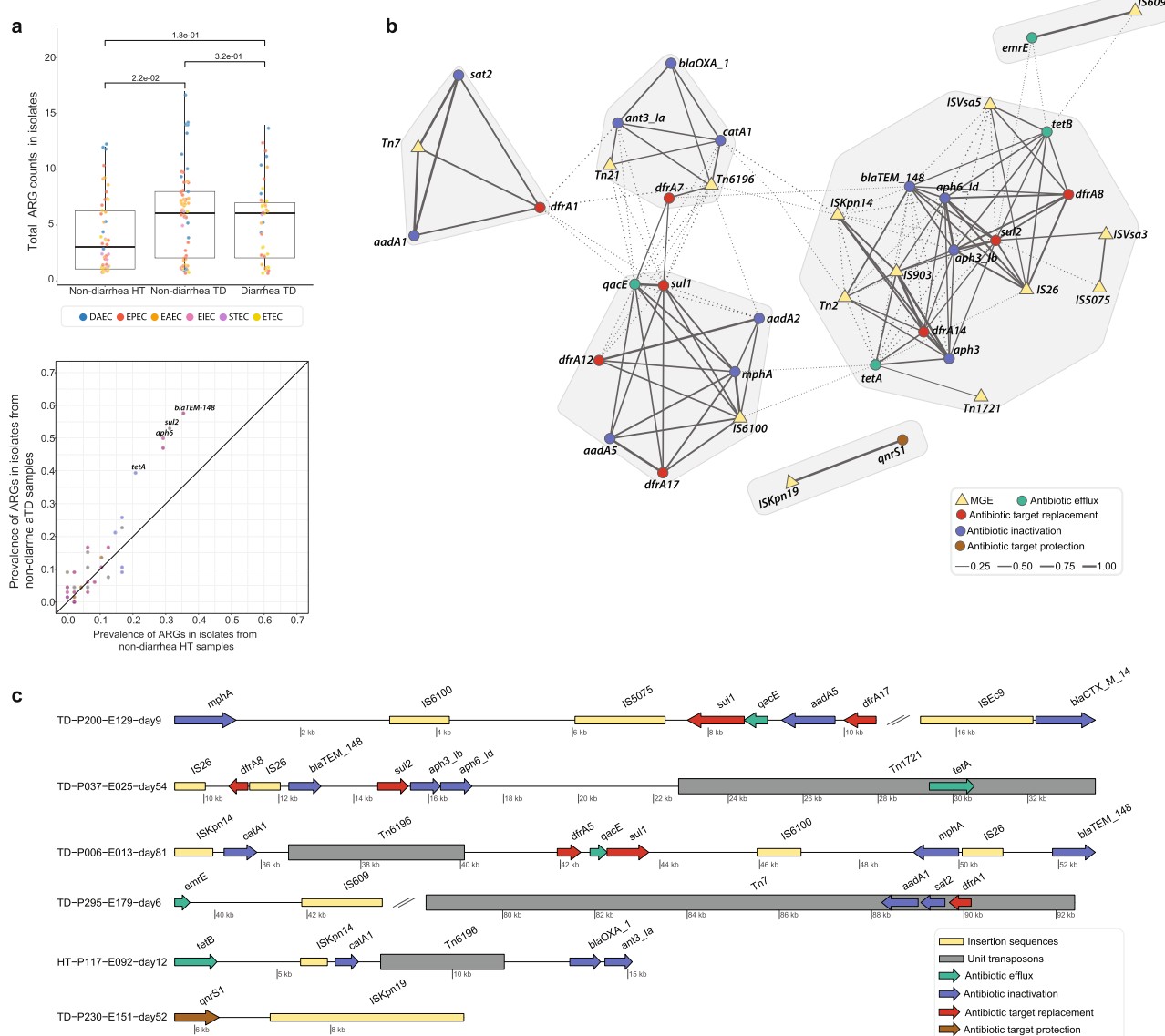

**Fig. 6 | ARG content of diarrheagenic *E. coli* isolates. a** Boxplot shows higher number of unique ARGs carried by DEC strains in non-diarrhea samples of TD subjects compared to HT subjects (*n* = 189, Wilcoxon test, *P* < 0.05). The colored data points represent different DEC strain. *P* values are multiple-hypothesis test corrected using Benjamini–Hochberg (FDR) method. Boxes show median and quartiles; error bars extend to the values within 1.5 interquartile range. Dotplot at the bottom shows the prevalence of ARGs in DEC isolates obtained from non-diarrheal HT and non-diarrheal TD subjects' samples. The annotated genes (*blaTEM-148*, *sul2*, *aph6*, and *tetA*) were significantly associated with TD subjects after FDR correction (Fisher exact test; *P* < 0.05). **b** Co-occurrence network of ARGs (circles) and mobile elements (triangles). The edges between two nodes represents the correlation of ARGs and mobile elements in the same contig (within 5 kb). Solid lines represent significant relationships after FDR correction (Student's *t* test, *P* < 0.05) and shaded regions represent subnetworks. **c** Examples of arrangements of ARGs and mobile elements identified on the same contigs. Underlying data are provided in the Source Data file.

intervention studies focused on the impacts of specific treatment regimens and/or prophylaxes (e.g., prebiotics, probiotics, synbiotics) on the gut microbiome and resistome could further inform best practices to mitigate the collateral damage from antibiotic use[69–72]. While it is unlikely that all taxa and ARGs that correlate with travel and diarrhea outcomes in our study would translate to other countries, which could have different infectious diseases burdens and ARG abundances, we predict that our findings regarding microbiome and resistome dynamics and acquisition of MDROs may be broadly generalized. Further multi-cohort travelers' studies to countries with different pathogen etiologies are warranted to best inform treatment and mitigation strategies that minimize microbial disruption and MDRO acquisition during travel.

## Methods

### Study design and cohort overview
Our cohort of international travelers was comprised of Spanish language students who were enrolled at the Amauta Spanish School in Cusco, Peru. A clinic staffed by a study physician was established inside the school to enroll and follow subjects. Travelers aged ≥18 years, able to understand written English, and enrolled at the school as students were eligible. The majority of these travelers stayed in lodging provided by the school within its premises. The study physician explained the purpose and procedures of the study to potential participants within 24 to 48 h of their arrival in Cusco city. Those who agreed to participate signed an informed consent form and completed a standardized enrollment questionnaire, that collected baseline

demographic and epidemiologic data, as well as medical and travel history (Form A in Supplemental Information). Subjects were also provided a stool collection kit and instructions to collect a baseline stool sample with their next bowel movement. Weekly stool samples were collected from subjects throughout their stay, and a weekly questionnaire collected data on unreported diarrheal episodes and dietary habits. Subjects were instructed to immediately communicate with the study physician when having a diarrheal episode and before taking any medication (e.g., antibiotics). Diarrhea was defined as passing one or more semi-liquid or watery bowel movements associated with gastrointestinal symptoms such as abdominal pain, cramping and/or nausea. Subjects reporting diarrhea were asked to provide an extra stool sample and were examined daily. During the diarrheal illness, 70 participants were prescribed antibiotics (e.g., ciprofloxacin, azithromycin) supplemented with oral rehydration therapy, and the remaining individuals were advised only oral rehydration therapy. Few individuals ($n = 8$) took antispasmodic and antiparasitic medication. Physical exam findings, symptoms, duration, and medications were documented in an acute diarrhea questionnaire (Form B in Supplemental Information). Diarrheal episodes were treated as independent events if an asymptomatic period of at least seven days occurred between them. Samples were classified as non-diarrhea or diarrhea based on whether they were collected during a diarrheal episode or not. Samples were also given a stool grade based on the Bristol stool form scale (BSS) of stool consistency, ranging from grade 1 (normal, hard) to grade 4 (loose, diarrheal).

The study protocol and its amendments were reviewed and approved by the Institutional Ethics Committee of the Universidad Peruana Cayateno Heredia, the Institutional Review Board of the US Naval Medical Research Unit No. 6 (NAMRU-6), and the Washington University in St. Louis Institutional Review Board.

## Empirical selection of diarrheal samples for analysis

Since diarrhea is a complex clinical infection and reliance on self-reported information alone can be misleading, we implemented a multi-factor selection procedure using epidemiological and clinical information to screen diarrheal samples prior to downstream analyses. First, we verified that the diarrheal samples were collected within the documented onset and recovery dates. If the recovery date was missing from the metadata, only diarrheal samples collected within two days of onset were included; if both onset and recovery dates were missing, then only samples that had a stool consistency of 3 or 4 (semi-liquid to liquid) were included. We then implemented a modified scoring scheme[73,74] referred here as the TD score, to further minimize the false positives (Supplementary Table 14). The TD score combined the presence of clinical signs (such as fever, vomiting, dehydration e.t.c.) and macroscopic data (stool consistency) collected during the diarrheal episode into a single metric. These parameters included: (1) stool frequency (in the last 24 h) and consistency; (2) duration of diarrhea; (3) participant-reported dehydration; (4) behavioral signs (e.g., nausea, fatigue, headache, bloating, anorexia) and/or clinical symptoms (e.g., temperature >38 °C); (5) pulse rate >100; and (6) presence of blood in stool. Each parameter was further divided into thirds and assigned equal points based on the severity of the symptom (1 point: bottom-third, 2 points: middle-third, and 3 points: upper-third) and were aggregated across parameters. We analyzed the TD score validity by comparing it with individual factors such as stool grade, maximum stool frequency in 24 h, and duration of diarrhea. While the individual factors did not necessarily correlate with each other, the TD score was significantly correlated with the given factors (Supplementary Discussion). Diarrheal samples with a TD score greater than 1.5 were included in the analysis. Using this multi-faceted filtering strategy, we selected 102 diarrheal samples total for downstream analyses.

## Stool specimen processing

All collected stool samples were delivered to the onsite clinic. Samples were refrigerated at 4 °C and preserved in Cary Blair media then transported to the Universidad Peruana Cayetano Heredia laboratory in Cusco, Peru within 2 h. Stool samples from diarrheal episodes that started during the night hours were collected by study personnel and refrigerated at 4 °C until they could be delivered the next morning. In the laboratory, stool samples were examined macroscopically for consistency and appearance, microscopically for parasite detection and fecal leukocytes, and underwent routine stool culture followed by antimicrobial sensitivity testing. Aliquots of fresh stool with no preservatives were frozen at −80 °C. Primary isolates and fresh stool aliquots were shipped in dry ice on a monthly basis to NAMRU-6 in Lima, Peru for further workup and long-term storage. Samples stored at −80 °C were shipped to Washington University in St. Louis, MO where the samples were stored at −80 °C until DNA extraction.

## Stool assessment and cultures for enteropathogen detection and antibiotic susceptibility testing of isolates

Stool samples were collected and processed immediately upon arrival at the nearby laboratory in Cusco. Gross examination and Hemoccult card tests (Beckman Coulter, Brea, CA) were used to evaluate for fecal gross and occult blood, respectively. Stool microscopy using direct iodine wet mount, modified Kinyoun stain, and methylene blue stain techniques were performed to examine for the presence of parasite ova, cyst, and fecal leukocytes, respectively as previously described[75]. Bacterial cultures were performed for the presence of pathogenic *E. coli*, *Salmonella* spp., *Shigella* spp., *Campylobacter* spp., *Aeromonas* spp., *Plesiomonas* spp., and *Vibrio* spp. All primary, enrichment, and biochemical differentiation culture media, BBL Sensi-Discs susceptibility discs, and catalase and oxidase test reagents were purchased from Difco BD Bioscience (Franklin Lakes, NJ). Fresh stool samples and stools preserved in Cary Blair transport media were cultured for bacterial enteropathogens using conventional microbiologic techniques[76,77]. Enteropathogens were identified according to their growth on differential and selective agar plates[78]. Stool specimens were streaked onto MacConkey agar, *Salmonella-Shigella* agar (SS), Hektoen Enteric agar (HE), thiosulfate citrate bile salt sucrose agar (TCBS), and *Campylobacter* blood-free agar base (CBF). All agar culture plates were incubated at 37 °C for 24 h after inoculation with the exemption of *Campylobacter* blood-free agar base plates that were incubated at 42 °C for 48 h in microaerophilic conditions. Stool samples were also inoculated into selenite enrichment broth and peptone water as sub-cultures and incubated at 37 °C for 18–20 h after inoculation. After incubation, the selenite enrichment broth was streaked onto HE agar to assess for *Shigella* spp. and the peptone water was streaked in the TCBS agar to screen for *Vibrio* spp. and both were incubated at 37 °C for 24 h. Five lactose fermenting colonies with morphology compatible with *E. coli* were selected from each MacConkey agar plate and preserved for polymerase chain reaction detection of pathogenic *E. coli* as described below. Other colonies with morphology and characteristics of enteropathogens seen on the MacConkey, SS, HE, and TCBS agars were inoculated in motility indole ornithine agar (MIO), Kliger iron agar (KIA), lysine iron agar (LIA), and Simmons citrate (CIT) and incubated at 37 °C for 24 h. After incubation, enteropathogen differentiation was performed according to their growth characteristics in the different agars and differential biochemical reactions[78]. Colonies growing on the CBF agar were evaluated with oxidase, catalase, and gram stain testing to differentiate *Campylobacter* colonies. Colonies with morphology and biochemical profile consistent with *Shigella* were confirmed by agglutination with serotype-specific antisera.

After biochemical identification, all the lactose fermenting and enteropathogens colonies identified were shipped to NAMRU-6 in Lima for quality control and further workup. For transportation,

lactose fermenting colonies were inoculated again in MacConkey agar and *Campylobacter* colonies were inoculated in CBF agar, all were incubated at 37 °C for 24 h, and isolated colonies resuspended in cryovials with trypticase soy broth with 15% glycerol, frozen at −70 °C, and shipped overnight on dry ice. The colonies identified as *Salmonella* spp., *Shigella*, *Aeromonas* spp., *Plesiomonas* spp., and *Vibrio* spp. were inoculated in vials containing trypticase soy agar, incubated at 37 °C for 24 h, and shipped overnight at 4–8 °C.

All identified enteropathogen colonies were inoculated on Mueller-Hinton (MH) agar to perform antibiotic susceptibility testing using the Kirby Bauer disk diffusion method according to the Clinical and Laboratory Standards Institute (CLSI) guidelines[79]. Antibiotic susceptibility testing was performed with 17 antibiotic discs (amoxicillin-clavulanate, ampicillin, azithromycin, ceftriaxone, chloramphenicol, cephalothin, cefepime, ciprofloxacin, rifaximin, amikacin, gentamicin, imipenem, tetracycline, trimethoprim-sulfamethoxazole, ticarcillin, ticarcillin/clavulanate, and furazolidone). Colonies were suspended in sterile saline and turbidity adjusted to the 0.5 McFarland standard before plating onto MH agar using the lawn streak technique. Antibiotic discs were added with proper spacing between and plates were incubated at 37 °C for 18–20 h after inoculation. Zones of inhibition were measured and recorded as the diameter of growth inhibition, with interpretation of susceptibility or resistance determined using CLSI M100 published breakpoints. It is important to note that currently there is no established CLSI breakpoint for determining the azithromycin resistance in *E. coli*. However, azithromycin has been found to be highly effective and is frequently used in treating diarrheal infections caused by Gram-negative pathogens, including diarrheagenic *E. coli*. In this study, we defined azithromycin resistance as per[80], where the minimum inhibitory concentration of azithromycin was determined in accordance with CLSI guidelines using the agar dilution method on all isolates with halo diameter <15 mm.

**PCR detection of pathogenic *E. coli*.** Five lactose-positive colonies were selected randomly from each participant stool culture plate and shipped frozen at −70 °C to NAMRU-6 in Lima for quality control and real-time florescence-based multiplex PCR for the detection of the currently recognized classes of diarrheagenic *E. coli* as previously described[81,82].

Frozen stocks of isolated colonies were restreaked onto MacConkey agar and incubated at 37 °C for 18–20 h. For DNA extraction, a single lactose-positive colony was carefully removed from the MacConkey agar plate using a sterile toothpick to avoid agar contamination and placed individually into a vial with 100 µl of sterile molecular-grade water. DNA was extracted by boiling at 100 °C for 5 min, then 5 min on ice, followed by centrifugation at 14,000 rpm for 10 min at 4 °C.

Primers were designed as previously described[82] to detect eight different virulence genes simultaneously in a single reaction, including aggR for enteroaggregative *E. coli* (EAEC), ST1a/ST1b, and LT for enterotoxigenic *E. coli* (ETEC), eaeA for enteropathogenic *E. coli* (EPEC), stx1 and stx2 for Shiga toxin-producing *E. coli* (STEC), ipaH for enteroinvasive *E. coli* (EIEC), and daaD for diffusely adherent *E. coli* (DAEC)[82]. The primers were designed so that amplicons had differential melting temperatures (TmS) ranging from 77 to 95 °C. The primers were diluted with TE buffer pH: 8.0 to obtain a 100 µM stock concentration, then further diluted to 25 µM working concentration with molecular grade water.

Two microliters of the DNA extraction lysate was used as a template added with 23 µl PCR master mix for a final reaction volume of 25 µl including Phusion High Fidelity buffer (Finnzyme OY, Espoo, Finland) and 0.5 U Phusion polymerase with 200 µM deoxynucleoside triphosphates and 4 mM MgCl$_2$ (ThermoScientific, Waltham, MA). Sybr green I was added as recommended by the manufacturer (Cambrex Bio Science, Rockland, ME). The qPCR was performed on Rotor-Gene Q PCR thermocycler with software version 1.7 (Qiagen, Germantown MD). Cycling conditions were 98 °C for 50 s, 60 °C for 20 s, 72 °C for 30 s, and 75 °C for 1 s over a total of 25 cycles, after which melting curves were determined with a ramp speed of 2.5 °C/s reading each 0.2 °C increments between 73 and 95 °C[82]. Melting peaks were calculated by the software to allow the interpretation of reactivity for the different virulence factor genes. Reference strains for each pathogenic *E. coli* were included as positive controls in the reaction.

**RT-PCR for norovirus detection.** Stool specimens were transported in portable coolers on ice packs to the field laboratory, aliquoted, and were stored at −80 °C until they were shipped in batches on dry ice to the NAMRU-6 laboratory in Lima for testing. Specimens were treated as described previously[83] with some modifications. In brief, fecal samples were diluted 10% in 1× phosphate-buffered saline, vortexed, and centrifuged at 5943 × g for 10 min at 4 °C[84]. Total RNA was extracted from stool-diluted samples using the QIAmp Viral RNA mini kit (Qiagen, Valencia, California) as described by the manufacturer. RNA was eluted with 60 µL of 0.01% of RNAse inhibitor (Qiagen) in DEPC Treated Water (Invitrogen, Life Technologies, Waltham, MA). RNA samples were stored at −80 °C until use. Norovirus detection was performed using a Duplex genogroup-specific real-time reverse-transcription polymerase chain reaction (qRT-PCR) developed and described by the National Calicivirus Laboratory at the Centers for Disease Control and Prevention in a 7500 FAST real-time platform (Applied Biosystems, Waltham, MA)[85,86]. RT-PCR was performed at 45 °C for 10 min and 95 °C for 10 min followed by 45 cycles of 15 seconds at 95 °C and 1 min at 60 °C using primers and probes from previous assays[87,88]. A sample was considered positive for norovirus GI when the calculated cycle threshold (Ct) was less than or equal to 37 cycles and considered positive for GII when the Ct was less than or equal to 39 cycles. Test results were only considered valid when all quality control positive reactions were below Ct cutoff and did not exhibit fluorescence above the threshold for the negative template control reactions.

## Metagenomic DNA extraction and sequencing

Metagenomic DNA (mgDNA) was extracted from 300–400 mg stool samples using the phenol-chloroform repeated bead-beating protocol as described previously[21]. The extracted mgDNA was diluted to 0.5 ng/µL and sequencing libraries were prepared using a modified Nextera protocol[89]. The libraries were purified using the Agencourt AMPure XP system (Beckman Coulter) and quantified using the Quant-iT Pico-Green dsDNA assay (Invitrogen). For each sequencing lane, 10 nM of approximately 96 samples were pooled three independent times. These pools were quantified using the Qubit® dsDNA BR Assay and combined in an equimolar fashion. Samples were submitted for 2x150bp paired-end sequencing on an Illumina NextSeq-High Output platform at The Edison Family Center for Genome Sciences & Systems Biology at Washington University in St. Louis.

## Metagenome profiling

Metagenomics raw reads were processed to remove Illumina adapter/index sequences and low-quality reads using Trimmomatic (v0.36)[90] with the following parameters ILLUMINACLIP:NexteraPE-PE.fa:2:30:10:1:true; SLIDINGWINDOW:4:15; LEADING:10 TRAILING:10; MINLEN: 60. The removal of human reads contaminants was performed using deconseq (v0.4.3)[91]. Post-cleaning, 710 samples with >3 million reads were used in downstream analyses. The taxonomic composition was determined using Metaphlan2[17]. Taxa with <0.01% relative abundance in >90% of samples were removed, resulting in 143 species included in downstream comparative analyses.

## Functional metagenomics

We constructed 21 small-insert (2–5 kb) functional metagenomic libraries, which were screened against 17 antibiotics (Supplementary

Data 1) based on our previously published protocols[19–23]. The experimental protocol is described below:

## Library preparation

The mgDNA extracted from 10 randomly selected stool samples were pooled together for the construction of each individual functional metagenomic library. The pooled mgDNA was sheared with Covaris E220 sonicator following the manufacturer's recommended settings for a target size of 3 kb (DNA mass: 2–20 µg of mgDNA diluted in 200ul of Buffer EB, intensity: 0.1, duty cycle: 20%, cycles per burst: 1000, treatment time: 600 sec). The sheared mgDNA was size-selected by gel-electrophoresis (1% agarose, 0.5× Tris-borate-EDTA, GelGreen dye (Biotium), 70 V for 130 min). The band covering 2–5 kb fragment size was excised, then mgDNA fragments were recovered using the QIAquick gel extraction kit (Qiagen). The size-selected mgDNA fragments were end-repaired with the END-It™ DNA End Repair kit (Epicenter) and purified with the QIAquick PCR purification kit (Qiagen). The end-repaired mgDNA was quantified with the Qubit HS fluorometer assay kit (Invitrogen) and concentrated using a vacuum concentrator (SpeedVac®) to a final volume of ~10 µl. The mgDNA fragments were then ligated into the pZE21-MCS-1 vector at the *BamHI* site. Linearization of the pZE21 vector was performed by inverse PCR using the following reaction conditions: (1) Mix 10 µl of 10× reaction buffer, 1.5 µl of 10 mM dNTP mix, 1 µl of MgSO4, 1 µl of 100 pg/µl circular pZE21 plasmid vector, 0.75 µl forward primers (5′ GACGGTATCGATAAGCTTGAT 3′), 0.75 µl reverse primers (5′ GACCTCGAGGGGGGGG 3′), 0.4 µl blunt-end HF DNA polymerase and 29.6 µl of nuclease-free water to a final volume of 50 µl. (2) PCR settings: 95 °C for 5 min, then 35 cycles of [95 °C for 45 sec, 55 °C for 45 sec, 72 °C for 2.5 min], then 72 °C for 5 min. The linearized pZE21 plasmid vector was size-selected (~2200 bp) using gel-electrophoresis, purified with the QIAquick PCR purification kit (Qiagen), and then dephosphorylated with calf-intestinal alkaline phosphatase (CIP, New England BioLabs) by adding 40 µl of gel-purified DNA, 5 µl of CIP (10 U/µl), and 5 µl of the 10× reaction buffer. The 50 µl reaction was incubated overnight at 37 °C before the CIP was heat-inactivated by incubating the reaction mix at 70 °C for 15 min. The ligated plasmid DNA was then purified using the QIAquick PCR purification kit (Qiagen) dialyzed with cellulose membrane (Millipore, VSWP09025) for 30 min, and then electroporated into *E. coli* MegaX DH10B (Invitrogen) following the manufacturer's instructions. After transformation, cells were recovered in 1 ml of recovery medium (Invitrogen) for 1 h at 37 °C. The library titers were determined by plating 0.1 µl and 0.01 µl of recovered cells onto Luria-Broth (LB) agar plates containing 50 µg/ml kanamycin as previously described[20]. The remainder of recovered cells were grown overnight in 50 ml of LB broth containing 50 µg/ml kanamycin (LB-Kan). The culture was then centrifuged and resuspended in 15 ml of LB-Kan broth containing 15% glycerol and stored at −80 °C for subsequent screening. *Functional screening of antibiotic resistance:* Each metagenomic expression library was screened on Mueller-Hinton agar with 50 µg/ml kanamycin and one of the 17 antibiotics at concentration listed in Supplementary Data 1. Before plating each library on antibiotic-containing growth media, the concentration of each library was adjusted such that 100 µl of library freezer stock contained at least 10× the total number of unique clones as determined at the time of library creation. To adjust the concentration, the freezer stock solution was either diluted with MH-Kan or centrifuged and reconstituted again in the appropriate volume for plating. The antibiotic selection plates were incubated for 24 h at 37 °C to allow growth of antibiotic-resistant clones. Additionally, for each antibiotic selection, a negative control plate of *E. coli* MegaX DH10B strain transformed with unmodified pZE21 (i.e., without a metagenomic insert) was plated to ensure that the concentration of antibiotic used entirely inhibited the growth of clones containing unmodified pZE21. The surviving colonies from each antibiotic selection were collected by adding 750 µl of LB-Kan with 15% glycerol, and colonies were scraped with an L-shaped spreader. The slurry of antibiotic-resistant clones removed from the surface of the plate was stored at −80 °C before sequencing them with Illumina NextSeq platform.

## Sequencing, assembly, and annotation of functionally-selected resistance determinants

The plasmid DNA containing antibiotic-resistant mgDNA fragments was extracted from functionally-selected clones using the QIAprep Spin Miniprep kit (Qiagen), and prepared for sequencing with a Nextera protocol, as described above. The samples were submitted for sequencing using an Illumina NextSeq platform (2 × 151 bp paired-end reads). Reads from each antibiotic selection were assembled into contigs using PARFuMS[19], a tool specifically designed for high-throughput assembly of resistance-conferring DNA fragments from functional selections. Selections were excluded from analysis if: (1) the number of contigs assembled was 10 times more than the total number of colonies; or (2) >200 contigs were assembled. Contigs were also filtered based on length (>500 bp). A total of 7020 contigs were obtained, and 16,334 open reading frames (ORF) were predicted in these contigs using the gene-finding algorithm Prodigal[90]. These ORFs were then annotated using an in-house pipeline called *resAnnotator.py*, which follows a hierarchical approach: (1) ORFs are searched against BLAST-based ARG databases (CARD[46], ResFinder[46], and AMRFinder-Prot[47]) with high percent-identity (>95%) and coverage (>95%); and (2) the remaining ORFs are annotated using HMM-based ARG databases (Resfams[91], AMRFinder-fam[47]). Overall, 1233 unique ARGs were identified.

## Resistome profiling of metagenomics samples

The relative abundance of ARGs in the metagenomics samples was calculated using ShortBRED[18](v0.9.4). First, we built a high-precision ARG-specific markers database from 7,921 antibiotic resistance proteins that were used as proteins of interest for identification of marker families using *'shortbred_identify.py'* with the following non-default parameters: --clustid 0.95 --ref Uniref90[92]. The antibiotic resistance protein sequences include sequences from the CARD database[46], the NCBI-AMR database[47], and antibiotic resistance proteins identified using functional metagenomics in this cohort, and previous studies[19,21–23,93–97]. In total, we generated a comprehensive set of markers database consisting of 6,585 unique marker sequences representing 2,331 AMR gene families. These AMR gene families were then manually curated, and entries with the following criteria were removed from analysis consideration because they would not be confidently expected to provide resistance based solely on a short-read marker (e.g., when that gene would require other components to provide phenotypic resistance, or when short-read markers would not distinguish between susceptible vs. resistant versions of an antibiotic target):

(1) Genes associated with global gene regulators, two-component system proteins, and signaling mediators (e.g., blaZ, vanS-vanR, mecI, mepR, gadW, marR);

(2) Genes encoding subunits that are part of multiple efflux pumps (e.g., tolC, oprM, opmD);

(3) Resistance via mutation in genes (e.g., resistance to antifolate drugs via mutations in dhfr, resistance to rifamycin via mutation in rpoB);

(4) Genes conferring resistance by modifying cell wall charge (e.g., mprF);

(5) Genes that reduce permeability (e.g., omp38, tmrB) or confer resistance through overexpression (e.g., Thymidylate synthase); and

(6) General efflux pumps that came through functional selections (e.g., MFS-type, ABC-type)

The relative abundance of AMR gene families was quantified by mapping reads to the filtered set of marker sequences using

*'shortbred_quantify.py'*. ShortBRED hits were filtered-out if they had counts lower than 2, or a mean reads per kilobase million (RPKM) lower than 0.001.

## Bacterial DNA isolation, sequencing, and assembly

The *E. coli* isolates stored at −80 °C in 15% glycerol were inoculated in 1.5 ml tryptic soy broth and were grown overnight at 37 °C with shaking. Genomic DNA was extracted using the Biostic Bactermia DNA isolation kit as per the manufacturer's protocol. DNA concentration was quantified using the Qubit fluorometer and stored at −20 °C. Isolate sequencing libraries were prepared using the Nextera protocol[89], and were sequenced on the Illumina NextSeq platform with 2 × 150 paired-end reads. Raw sequencing reads were binned by index sequences, quality filtered using Trimmomatic(v0.36)[98] and human contaminants were removed by DeconSeq (v0.4.3)[99]. Processed reads were assembled into draft genomes using SPAdes (v3.11.0)[100] with the following parameters: *spades.py -k 21,33,55,77 --careful*. Assembly quality was assessed using CheckM[101] and the following inclusion criteria were used for downstream analysis: >90% completeness, <5% contamination, and <500 contigs longer than 500 bp.

## Genomic analysis of *E. coli* isolates

The assemblies of 189 *E.coli* isolates and 40 publicly available *E.coli* reference genomes were annotated using Prokka (v1.12)[102] with default parameters. Multilocus sequence types and serotype were determined using MLST (https://github.com/tseemann/mlst) and SerotypeFinder[103].

In silico detection of antibiotic resistance and virulence genes in the isolate genomes was performed using the in-house pipeline *resAnnotator.py*, which sequentially annotates ORFs by searching against the given Blast- and HMM-based databases. For antibiotic resistance genes, we screened against the following databases in order: Resfinder[104], NCBI-AMR[47], CARD[46], and Resfams[91]. For virulence gene identification, VirulenceFinder[105,106] and VFDB[107] were used. Mobile genetic elements in the isolate genomes were predicted by MGEFinder[54]. Average nucleotide identities among isolates were computed using pyani (https://github.com/widdowquinn/pyani). The pangenome analysis was performed using Roary[49], with core-genome alignments created from assembled contigs of all *E. coli* isolates (189 sequenced and 40 published reference genomes). The reference genome of *E. marmotae* was also included as an outgroup. The maximum-likelihood core-genome phylogenetic tree was constructed using RAXML (v8.2.11)[50] with the following parameters: *-m GTRGAMMA -f a -N 1000 -x 25418* and visualized using iTOL[108].

## Statistical analysis

**Identification of covariates.** To identify the metadata variables that significantly affected the taxonomic profile, functional profile, and resistome profile, we calculated the total variance contribution of each variable using PERMANOVA with repeated measures, as described previously[25]. To account for repeated measurement in metadata variables that change over time (e.g., stool consistency, sample type, pathogen presence), permutations were limited to within the subjects. Meanwhile, variables that were constant across samples from the same subject (e.g., age, sex, travel history, lodging, country of origin) were first permuted across subjects, with samples re-labeled with the variable from their permuted subject. The significance value for metadata variables was determined using 5,000 permutations and p values that were FDR corrected for multiple-hypothesis testing. The metadata variables that showed significant variation among these profiles were included in the models as covariates.

**Longitudinal changes in alpha- and beta-diversity.** To model the effect of length of travel and diarrhea on gut microbial and AR gene diversity, a linear mixed effect model (LMM) (for modeling continuous response variables, e.g., Shannon index) or generalized linear mixed

effect model (GLMM) with Poisson family (for modeling count data e.g. richness) was fit by maximum-likelihood using lmer or glmer function in R, respectively. These models include regression on alpha diversity, beta-diversity, and cumulative change in abundance measures against diarrhea and time while also adjusting for the effects of age, sex, region, and inter-individual variability by specifying subjects as the random effects. Pseudo-R2 was determined using r.squaredGLMM function in the MuMin package. Post hoc pairwise comparison between different Sample Type (e.g., DiarrheaTD vs Non-diarrhea PostTD) was performed on fitted models using emmeans package and *P* values were corrected for multiple hypotheses using Benjamini−Hochberg method (FDR).

**Microbial features association analysis.** To identify microbial features (taxa or ARGs) associated with diarrhea, we performed multi-variable association analysis using MaAsLin2[27] that accounts for multiple covariates and repeated measures. Briefly, Masslin2, first applies an appropriate transformation/normalization method: arcsine square-root transformation for taxonomic profile and log transformation with half the minimum relative abundance as a pseudo count. Then, the transformed abundances of each feature were fit with the following linear mixed-effects model:

$$microbial\,feature\,abundances\,(taxa\,or\,AR\,genes) \sim Sample\_type + Age + Sex + Region + AbxUse + Time\,in\,Peru\,(days) + 1|Subjects)$$

where each microbial feature was modeled as a function of diarrhea while adjusting for the effect of metadata variables (significant covariates that came from PERMANOVA analysis) as well as accounting for inter-individual variability by specifying the subjects as random effects. Further, *p* values of each feature were corrected for multiple-hypothesis testing using Benjamini−Hochberg method with FDR < 0.25.

## Microbiome network analysis

We inferred two unsupervised co-occurrence networks from diarrhea (case) and non-diarrhea (control) samples at the species-level using the SparCC[44] algorithm which calculates the correlations between microbial species while taking into account the sparsity and inherent compositionality in the microbe relative abundance data. To account for differences between the number of diarrheal and non-diarrheal samples, we took subset of samples to construct co-occurrence networks. For the diarrhea network, only the first diarrhea samples per individual were considered, whereas non-diarrheal samples that were collected within the first two weeks of stay and were not flanked by a diarrheal episode were included in the non-diarrhea network. For the given diarrhea and non-diarrhea datasets, SparCC was used with default parameters to calculate the correlations among species, and then the "*MakeBootStrap*" command was applied to generate 100 bootstrap tables, which were again used to calculate the SparCC correlations. Finally, the bootstrapped correlations were used with the "*PseudoPvals.py*" command to generate two-tailed p values from the true table. The correlation values with *p* values <0.05 were retained. We then compared the two co-occurrence networks using NetShift[45] which quantifies the changes in the interactions of the individual node to identify the "driver taxa". The key parameters for identifying the driver taxa were the NESH score, Jacard index and delta betweenness centrality. The co-occurrence networks were drawn using the *igraph* package in R.

## Microbiome shift events

Microbiome "shift"[25] events were defined as when the Bray−Curtis dissimilarity index between two consecutive samples from a single individual was more likely to have come from the distribution of the Bray−Curtis dissimilarity index derived from samples of different

individuals. Based on this definition, we first obtained the distribution of the Bray–Curtis dissimilarity index between samples from different individuals of the HT cohort and the Bray–Curtis dissimilarity index between samples from the same individuals of the HT cohort. The point at which the inter-individual dissimilarity estimate exceeded the intra-individual dissimilarity estimate was chosen as the threshold to define the "microbiome shift" event (Bray–Curtis dissimilarity: 0.52).

### Machine learning classification model

To distinguish diarrheal from non-diarrhea samples based on taxonomic composition, we built machine learning classification models using SIAMCAT[43], an ML-based tool box developed specifically for metagenomics studies[109]. The SIAMCAT[43] workflow consists of normalization of taxonomic features, splitting the datasets using cross-validation schemes (fivefolds), training with different machine learning models (LASSO logistic regression, Random Forest, and Elastic Net), and evaluating the performance of the models using area under the receiver operating characteristic curve.

### Reporting summary

Further information on research design is available in the Nature Portfolio Reporting Summary linked to this article.

## Data availability

All data from shotgun metagenomics, isolates, and functional metagenomics sequencing are available from the NCBI SRA under BioProject ID PRJNA698223. The source data underlying main text figures and supplementary figures are provided as a Source Data file. Source data are provided with this paper.

## Code availability

The assembly (PARFuMS[https://zenodo.org/badge/latestdoi/89020730]) and annotation (resAnnotator[https://zenodo.org/badge/latestdoi/331011517]) pipelines used in functional metagenomics are available on github at https://github.com/dantaslab. Analysis scripts used here are available from the authors upon reasonable request.

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

## Acknowledgements

The authors would like to thank Maria Silva, Yocelinda Meza, Maruja Bernal, Nilda Gadea, and Enrique Canal at the Naval Medical Research Unit No. 6 in Peru who assisted with the collection and shipment of samples. We would like to thank Pablo Tsukayama for logistical support with sample access and helping with sample processing in the initial phase of the study as well as Emily Deichsel who helped with participant enrollment and data collection. We also thank the staff at The Edison Family Center for Genome Sciences & Systems Biology at the Washington University School of Medicine in St. Louis, including Eric Martin and Brian Koebbe for computational support, Jessica Hoisington-López and MariaLynn Crosby for managing the high-throughput sequencing core, and Bonnie Dee, Kathleen Matheny, and Keith Page for administrative support. Finally, we would like to thank the members of the Dantas lab for helpful general discussions and comments on the manuscript. Disclaimers: the views expressed in this article reflect the

results of research conducted by the authors and do not necessarily reflect the official policy or position of the Henry M. Jackson Foundation for the Advancement of Military Medicine, Inc., Department of the Navy, Department of Defense, nor the United States Government. The study protocol was approved by the Institutional Review Boards at the Universidad Peruana Cayateno Heredia, the Naval Medical Research Unit No. 6, and the Washington University in St. Louis in compliance with all applicable federal regulations governing the protection of human subjects. Authors are military service members or federal/contracted employees of the US Government. This work was prepared as part of official duties. Title 17 USC § 105 provides that 'Copyright protection under this title is not available for any work of the US Government.' Title 17 USC § 105 defines a US Government work as work prepared by a military service member or employee of the US Government as part of that person's official duties. Funding: This work was funded by the Department of Defense Global Emerging Infections Surveillance (GEIS; funding number: CO693_12_LI), and the Congressionally Directed Medical Research Program through the Peer Reviewed Medical Research Program by an award to G.D. and M.P.S. (PRMRP; award number: PR170802). G.D. is also supported by the National Institute of Allergy and Infectious Diseases of the National Institutes of Health (R01-AI123394), and the Edward Mallinckrodt, Jr. Foundation (Scholar Award). K.S.B. is supported by the National Institute of Diabetes and Digestive and Kidney Diseases (T32-DK007130), and D.J.S is supported by Doris Duke Charitable Foundation Physician Scientist Fellowship (2021081) and NIH NIAID Clinical Scientist Research Career Development Award (K08-AI159384). The content is solely the responsibility of the authors and does not necessarily represent the official views of the funding agencies.

## Author contributions

G.D., M.P.S., D.H.T., and M.B. conceived the study design, experiments, and analysis. D.H.T., M.M.C., and M.P.S. assembled the cohort and oversaw sample and metadata collection. M.L.M., R.M., and G.S. led subject enrollment, sample and data collection from participants, and primary laboratory workup of collected samples. S.P. and M.B. extracted DNA from stool samples and isolates, and generated functional metagenomics libraries and shotgun data. M.B. performed the computational analysis and interpreted the results. M.B. and K.S.B drafted the article, with critical revisions from D.J.S., G.D., M.P.S., S.D.I., C.K.P., and co-authors. All authors reviewed and approved the final manuscript.

## Competing interests

The authors declare no competing interests.
