## [Peer Review File · Nature Communications]

REVIEWER COMMENTS

Reviewer #1 (Remarks to the Author):

This is a well-written report of the microbiome and resistome of longitudinally sampled travelers during travel to Cusco, Peru. The majority of studies of microbiome/resistome in travelers have been pre- and post-travel, and thus this study contributes a substantial amount of new and unique data with its “during-travel” sampling. This destination-based study minimizes the effects of travel destination, and sophisticated computational methods allow for very insightful conclusions. They report that traveler microbiotas do drift along the time of travel, but maintain diversity, and that post-diarrhea samples have decreased diversity and temporary disruption of the structure during the diarrhea. They importantly find that antimicrobial resistance genes are increased in travelers with diarrhea and remain elevated. The microbiologic methods are extensive (including culture- and genomics-based approaches), informatics is solid, but the analysis has several weaknesses as detailed below, including the lack of differentiation between TD treated w/antibiotics (ABx) and those untreated, which makes teasing out the contribution of diarrhea separate from antibiotics difficult. Nevertheless, given novelties outlined above, I recommend its publication if the major points could be addressed.

Major points:

-Diarrhea is suggested to be driving many observations- changes in alpha diversity, beta diversity, resistome, etc. However 70/113 of the donors with diarrhea took antibiotics. This is not accounted for in many of the analyses and should be considered before making these inferences. Specifically, analyses involving TD (e.g. Fig 1) should distinguish between TD treated w/ABx and untreated TD. To isolate the impact of diarrhea alone, the primary analysis should focus on only those who had TD without ABx treatment.

-I could not find any documentation of use of other medications that could potentially affect microbial diversity (such as bismuth use, or antimalarials, as some travelers may be visiting the Peruvian Amazon on the same trip). These should be detailed and accounted for in the analysis as well.

-Figure 1: “non-diarrhea TD” represents non-diarrheal stools from subjects who experienced TD. This is problematic, and there should be differentiation between “non-diarrhea pre-TD” and “non-diarrhea post-TD”. As shown in Fig 1b, the episode of TD changes the composition of the microbiome, and thus combining pre- and post-TD samples in the same group introduces a major confounder.

-Lines 254: discrimination between diarrheal and non-diarrheal samples are of unclear clinical relevance. Perhaps more relevant would be to examine the discrimination of microbiome of baseline samples to predict who ends up with TD and who does not.

Minor:

-Line 75: reference needed for, "diarrhea is predominately caused by bacterial etiologies,"

-Line 78-81: instead of ref 4, should use ref 3.

-Line 121: I could not locate where in Ext figure 2 that it show 69% get diarrhea in 1 month.

-Line 126, Excel table 2 is Diarrhea vs Non-Diarrhea, and many of the variables included in the analyses between HT vs. TD in Excel Table 1 is not replicated in Table 2 (e.g. most food data in excel table 2 is missing)

-Ext data figure 2e: Authors should specifically state that the percentages correspond to percent resistant

-Line 159 and figure 1b: Is decrease in diversity after diarrhea due specifically to changes in richness and/or evenness? E.g. are the communities more even after diarrhea?

-Figure 1a legend: to be consistent with other legends, should offer an explanation of the figure, instead of interpretation. Are the Shannon/Bray Curtis measurements from reads collapsed at a specific taxonomic level? (This applies to several other subfigures) Figure 1 or supplement should also show richness, before, during and after diarrhea

-line 164 (fig 1c): The rationale behind this analysis is not clear, as fig 1b already shows decreased diversity following TD. Instead, along with the statement made in the sentence following, the analysis should be to look at association of baseline taxonomic diversity with occurrence of TD.

-Line 181, ext figure 3a. Showing bray Curtis dissimilarity values within vs between sample types instead of a PCOA plot would better demonstrate the relationship stated.

-Line 188, are microbiome shifts in TD samples still increased when known pathogen is removed from the sequencing? Are microbiome shifts increased in HT when colonized with specific pathogen (compared to uncolonized HT)? Can HT and TD all with a known pathogen be compared, as well? As above, would perform analysis on whether microbiome shifts are more common in TD samples in those who were not treated with Abx.

-Line 210 and ext figure 3d, is enrichment of proteobacteria simply due to colonization with a pathogenic proteobacterium

-Ext Figure 3d. What statistical methods were used? There is a general lack of explanation in the figure legends regarding statistical testing.

-Figure 1F, most of the figure is not addressed in the text or legend.

- Typo on line 229, “splanchnicus”
- Figure 2a legend, should the adjectives, “left” and “right” really refer to “top” and “bottom”?
- Figure 3 legend—type “a)” appears twice. B and C are potentially mislabeled as well
- Line 288, as above, please check to see if AMR diversity and abundance is higher in samples from individuals that did not use antibiotics
- Line 297 should be reworded. The change in Bray Curtis over course of the stay does not solely “indicates that travelers acquire and/or lose specific ARGs 297 during the course of their stay...” Bray Curtis could increase due to increased evenness in the starting collection of AMR genes
- Figure 3e; is this only in diarrhea samples? Text makes it appear that way, but figure legend doesn’t specify. Because proteobacteria levels return to normal after diarrhea, but AMR abundance stays elevated, this conflicts with the statement, “...indicating that the majority of AR determinants in diarrheal samples are likely concentrated within Enterobacteriaceae species.” To support the suggestion that AMR genes are concentrated in Enterobacteriaceae, it would be nice to see AMR abundance in HT with/without detected Enterobacteriaceae pathogen
- Line 319: ref 46 does not provide primary data showing that bacterial etiologies are predominant cause of TD.
- Line 322: how did you differentiate between cases of EIEC and Shigella?
- Ext Fig 7b and d are not mentioned in the text
- Line 352: What breakpoint was used for azithromycin resistance? I am not aware of a CLSI breakpoint for azithromycin and E. coli, and I am concerned that the reader would make incorrect conclusions about clinical efficacy related to these rates of “resistance”.
- Figure 5a legend has a typo “shows”. Please provide a key for the different colors. Does the increased ARG in non-diarrheal TD vs HT samples hold if you look at samples before diarrhea (or is it driven by samples after diarrhea)?
- line 420: defining “temporary” and “persistent” colonization must be made in light of a recent report (Kantele et al, Lancet Microbe, 2021) showing with daily sampling in international travelers that culture-based ESBL-PE colonization status varied on a day-to-day basis.
- Line 427: I think instead of “highlighted”, should be “font colored”.
- Ext Fig 9: This figure is hard to understand—if they key showed SNVs associated with the heatmap it might be easier to follow. Parts of this figure aren’t referenced in the text either. It seems like there are figures to characterize virulence factors that were not discussed in the main text
- Line 435—does the SNV stability decrease around diarrhea in TD? Perhaps the lack of stability is due to a more virulent strain establishing and/or replacing a more innocuous strain.

Reviewer #2 (Remarks to the Author):

Boolchandani et al. describe a longitudinal study with 159 international students traveling to Peru. The study describes how travel changes the gut microbiota composition and how traveler's diarrhea impacts the species abundances in the gut. The authors have also investigated the antibiotic resistance gene profile shifts that occur alongside the changes observed in the microbiota composition. The study uses functional metagenomics, shotgun metagenomics, and genomic sequencing of strain and has produced a wealth of information to support the paper's main findings. The study complements the previous literature on the effect of travel on the antibiotic resistome, as previous studies have been limited by only including pre-and post-travel stool samples.

General comments:

The research has been done meticulously using appropriate methods and statistical approaches, and the sampling has been well planned. The analyses support the results and conclusions well. The article's language does not need additional revising.

The effect of travel on the resistome of travelers has been studied before. It is for example known that travelling to risk areas increases the odds of acquiring MDR pathogens. You could emphasize even more clearly how the results of the longitudinal sampling in a large cohort and the detailed analyses in your manuscript add to the existing literature. Highlighting the "most citable" items and results in your manuscript would improve the readability and make the paper more focused.

Please review the manuscript critically and try to make the text more concise. Now there are a wealth of results reported that do not necessarily support the main message of the paper, which might be slightly lost from the readers. Some results could be moved to the supplement or left out.

Please include sample sizes, p-values, q-values, or FDR values when reporting results of statistical tests and the name of the test used in the results section. Also, include 95 % confidence intervals and R², or similar metrics whenever applicable.

More detailed comments:

Lines 203-209: I am having a hard time following the train of thought in these sentences. Could the authors please revise and maybe shorten the sentences or simplify the language to make this section more easily understandable?

Lines 217-218: Please report the p-value or FDR cut-off used to identify species which were significantly differentially abundant.

Figure 1f: Figure guide with $-\log(\text{qval}) * \text{sign}(\text{coef})$ needs spelling out of the abbreviations in the figure legend and some explanation in how to interpret the scale since this is a non-standard way of showing differential abundances. For example, what does -6 mean? Is it a q-value of 10^{-6} and a depleted species? Are all of these taxa shown significantly differentially abundance, and what is their q-value?

Figure 1f and also elsewhere: Please use either qvalue or FDR throughout the paper since they mean the same thing (or clarify how they differ from each other).

Figure 1g: Why is there a p-value in the figure and not FDR or q-value? What test was used to calculate this.

Text referencing Figure 1f and 1g: Please include the name of the test and p-value, q-value, or FDR value.

Figure 2a: It is challenging to compare the node sizes to each other. Please include a guide for the size of the node and use categories or transformations for sizes to make it easier to see. You could also omit the abundance information from the figure.

Figure 2 a: What are the right and left networks? Aren't the networks arranged top and bottom?

Line 243: Could you clarify how SparCC or Netshifft identifies which taxa are enriched between HT and TD subjects?

Figure 2 b: Please include a guide for the node sizes. Please spell out NESH score in the figure legend.

Lines 239-253: Could you briefly clarify how "gained/lost connectivity" and "gained interactions" are measured? Is it tested statistically somehow, or are you comparing the number of connections of a species?

Figure 2 c: Could you give an example of how to interpret the z-scores? If the z-score is -3, what does it mean?

Figure 3f: What does log normalized FDR value mean in the scale? How should I interpret value -5? Are these all significantly different, and what is the test? There are two panels within the f-panel, but only one of them is discussed in the figure legend. Please include text for the other panel (left side) or remove this part of 3f.

Figure 3g: Why are these p-values and not FDR values? Should these be corrected for multiple testing as there are several genes you were testing?

Lines 319- 391: Could this section or a part of it be shortened or added to supplemental results? This section has parts that are not crucial for understanding the main message of the manuscript, and the manuscript could be made more compact by moving some of the text to the supplement.

Lines 419: Why are you performing pairwise co-occurrence comparisons if you have genome assemblies? Isn't it possible to see from the contigs which ARGs co-occur? Or what does "pairwise co-occurrence comparison" mean in this context? Are you utilizing the assembled contigs? Please clarify.

Lines 1212-1260: Figure legend texts on these lines and below the figures themselves are different in at least their numbering. Please revise.

We thank the reviewers for their valuable feedback. Responses to reviewers are detailed in **blue** along with line numbers corresponding to the changes in the manuscript.

Reviewer #1 (Remarks to the Author):

This is a well-written report of the microbiome and resistome of longitudinally sampled travelers during travel to Cusco, Peru. The majority of studies of microbiome/resistome in travelers have been pre- and post-travel, and thus this study contributes a substantial amount of new and unique data with its “during-travel” sampling. This destination-based study minimizes the effects of travel destination, and sophisticated computational methods allow for very insightful conclusions. They report that traveler microbiotas do drift along the time of travel, but maintain diversity, and that post-diarrhea samples have decreased diversity and temporary disruption of the structure during the diarrhea. They importantly find that antimicrobial resistance genes are increased in travelers with diarrhea and remain elevated. The microbiologic methods are extensive (including culture- and genomics-based approaches), informatics is solid, but the analysis has several weaknesses as detailed below, including the lack of differentiation between TD treated w/antibiotics (ABx) and those untreated, which makes teasing out the contribution of diarrhea separate from antibiotics difficult. Nevertheless, given novelties outlined above, I recommend its publication if the major points could be addressed.

Response: We thank the reviewer for the accurate summary of our work, and recommendations for improvement. Below we have provided detailed responses for the major and minor points. Our manuscript is greatly strengthened and improved as a result of addressing reviewer comments and suggestions.

Major points:

- Diarrhea is suggested to be driving many observations- changes in alpha diversity, beta diversity, resistome, etc. However, 70/113 of the donors with diarrhea took antibiotics. This is not accounted for in many of the analyses and should be considered before making these inferences. Specifically, analyses involving TD (e.g., Fig 1) should distinguish between TD treated w/ABx and untreated TD. To isolate the impact of diarrhea alone, the primary analysis should focus on only those who had TD without ABx treatment.

Response: We agree with the reviewer that antibiotic treatments may confound the overall microbiome and resistome dynamics during international travel and/or during diarrheal episodes. To address this concern, we have updated all our analyses using linear mixed effect models (LME) models to account for antibiotic usage. This includes regression on alpha diversity, beta diversity, and cumulative change in abundance measures against diarrhea and time while also adjusting for the effects of age, gender, region, and inter-individual variability by specifying subjects as the random effects. Below is a sample equation for alpha-diversity:

Alpha-diversity (Richness or Shannon) ~ SampleType (DiarrheaTD, PreTD, PostTD, NondiarrheaHT) + AbxUse + Age + Gender + Region + TimeInPeru(days) + 1/Subjects

Using the above structure, we have also revised microbiome and resistome analyses to account for antibiotic usage and have updated the respective figures and tables. Additionally, we have updated the Methods section to describe these changes in alpha and beta diversity measures for microbiome and resistome. (Line 1141-1151)

- I could not find any documentation of use of other medications that could potentially affect microbial diversity (such as bismuth use, or antimalarials, as some travelers may be visiting the Peruvian Amazon on the same trip). These should be detailed and accounted for in the analysis as well.

Response: There were few diarrheal episodes (n=8) where individuals had taken antispasmodic or antiparasitic medications. In most cases, participants took antibiotics supplemented with oral rehydration salts (ORS) or just ORS. We have added this information in the supplementary notes and methods section (Line 819-823).

- Figure 1: “non-diarrhea TD” represents non-diarrheal stools from subjects who experienced TD. This is problematic, and there should be differentiation between “non-diarrhea pre-TD” and “non-diarrhea post-TD”. As shown in Fig 1b, the episode of TD changes the composition of the microbiome, and thus combining pre- and post-TD samples in the same group introduces a major confounder.

Response: We agree with reviewer’s concern that including pre- and post-TD samples in the same group while analyzing the changes in microbiome and resistome composition is a confounder. In this revised manuscript, we have subdivided the “non-diarrheal TD” group into “non-diarrheal PreTD” and “non-diarrheal PostTD” samples, with this change reflected in all downstream analyses and figures.

- Lines 254: discrimination between diarrheal and non-diarrheal samples are of unclear clinical relevance. Perhaps more relevant would be to examine the discrimination of microbiome of baseline samples to predict who ends up with TD and who does not.

Response: We agree with the reviewer that identification of discriminatory taxa that could predict whether an individual would get diarrhea would be of significant clinical relevance; however in the current study, we don’t have enough baseline samples from both groups to build a robust predictive model for this purpose. With the current analysis, we used machine learning to identify diarrhea-specific microbial “signatures”. These are then evaluated in the context of significantly-enriched taxa (using MaAsLin2) and interactions between species and overall changes in the gut architecture (using SparCC and Netshift network analyses). This analysis can be useful for future studies of other traveler cohorts to see if these signatures are generalizable or geography specific. We have reworded the text in the Results and Discussion sections to emphasize this point (Line 243-257).

Minor points:

- **Line 75:** reference needed for, “diarrhea is predominately caused by bacterial etiologies,”
Response: We have added the following reference “*Travelers' diarrhea: update on the incidence, etiology and risk in military and similar populations - 1990-2005 versus 2005-2015, does a decade make a difference?*”
- **Line 78-81:** instead of ref 4, should use ref 3.
Response: We apologize for this oversight, and have corrected the reference.
- **Line 121:** I could not locate where in Ext figure 2 that it shows 69% get diarrhea in 1 month.
Response: We apologize for this textual error, and have removed the reference from the main text.
- **Line 126:** Excel table 2 is Diarrhea vs Non-Diarrhea, and many of the variables included in the analyses between HT vs. TD in Excel Table 1 is not replicated in Table 2 (e.g. most food data in excel table 2 is missing)
Response: Extended Data Table 1 contains information collected from a questionnaire that was asked at the time of enrollment and completion of the study, whereas the information provided in the Extended Data Table 2 was collected during a weekly follow-up.
- **Line 159 and figure 1b:** Is decrease in diversity after diarrhea due specifically to changes in richness and/or evenness? E.g. are the communities more even after diarrhea?
Response: We find that both Richness and Shannon index significantly decrease after diarrhea, as shown in Extended Data Table 4b.
- **Line 164 (Fig 1c):** The rationale behind this analysis is not clear, as fig 1b already shows decreased diversity following TD. Instead, along with the statement made in the sentence following, the analysis should be to look at association of baseline taxonomic diversity with occurrence of TD.
Response: Fig 1b shows transient decrease in alpha-diversity two weeks after diarrheal episodes in just TD subjects, while Fig 1c uses beta-diversity to show how the microbial community diverges throughout the total length of travel in both HT and TD subjects.
- **Line 181, Extended figure 3a:** Showing bray Curtis dissimilarity values within vs between sample types instead of a PCOA plot would better demonstrate the relationship stated.
Response: We believe that PCoA analysis appropriately shows the heterogeneity among temporally matched TD samples during diarrheal episodes. It also shows the weak association between sample type, which wouldn't be possible just by comparing Bray-Curtis dissimilarity values within and between subjects' samples.

- **Line 188**, are microbiome shifts in TD samples still increased when known pathogen is removed from the sequencing? Are microbiome shifts increased in HT when colonized with specific pathogen (compared to uncolonized HT)? Can HT and TD all with a known pathogen be compared, as well? As above, would perform analysis on whether microbiome shifts are more common in TD samples in those who were not treated with Abx.

Response: Strain variability within species makes it difficult to differentiate between pathogenic vs. benign strains based on metagenomic DNA sequencing data alone, particularly in asymptomatic samples (e.g., unable to distinguish between commensal vs. diarrheagenic *E. coli*). Therefore, we believe it would be inappropriate to base analyses around the presence/absence of putative pathogenic species.

- **Line 210 and ext figure 3d**, is enrichment of proteobacteria simply due to colonization with a pathogenic proteobacterium

Response: At the phylum level, we saw an increased abundance of Proteobacteria in diarrhea samples compared to pre- and post- non-diarrheal samples (Extended Data Fig. 3d). As shown in Fig 1f, the enrichment of Proteobacteria was associated with both Proteobacteria spp. not associated with causing diarrhea (e.g., *Bilophila* spp., *Sutterella wadsworthensis*, *Parasutterella excrementihominis*), as well known diarrhea-causing pathogens (e.g., *E. coli*, *Shigella* spp.). Previous studies have associated these non-pathogenic taxa with inflammatory bowel diseases^{1,2} and colon cancer³, but the causative role of these taxa leading to host disease is not yet established. Additionally, as stated above, even within species known to cause diarrhea, high strain-level variability within these species makes it difficult to differentiate between pathogenic vs. benign strains.

- **Line 229:** Typo on line 229, “splanchnlcus”

Response: Thank you for catching this error. We have corrected the strain name “*splanchnlcus*” to “*splanchnicus*” (**Line 233**).

- **Line 288**, as above, please check to see if AMR diversity and abundance is higher in samples from individuals that did not use antibiotics

Response: Antibiotic use is associated with significantly higher ARG richness and abundance but not Shannon diversity (Extended Data Table 10b).

- **Line 297** should be reworded. The change in Bray Curtis over course of the stay does not solely “indicates that travelers acquire and/or lose specific ARGs during the course of their stay...” Bray Curtis could increase due to increased evenness in the starting collection of AMR genes

Response: Re-analysis using the updated subject groups requested by the reviewer rendered this comparison non-significant, so this interpretation has been removed.

- **Line 319:** ref 46 does not provide primary data showing that bacterial etiologies are predominant cause of TD.

Response: We have corrected the reference.

- **Line 322:** how did you differentiate between cases of EIEC and Shigella?
Response: As described in the methods, the differentiation between EIEC and Shigella was done by a combination of differential selection on agar plates (MacConkey, Salmonella-Shigella agars), biochemical ID on API-20E, and multiplex PCR⁴.
- **Line 352:** What breakpoint was used for azithromycin resistance? I am not aware of a CLSI breakpoint for azithromycin and *E. coli*, and I am concerned that the reader would make incorrect conclusions about clinical efficacy related to these rates of “resistance”.
Response: The reviewer is correct that at present there is no established CLSI breakpoints for determining the azithromycin resistance in *E. coli*. Nonetheless, azithromycin has been found to be highly effective and is frequently used in treating diarrheal infections caused by Gram-negative pathogens, including diarrheagenic *E. coli*, and so we felt its inclusion was important. In this study, we defined azithromycin resistance as per ⁵, where the minimum inhibitory concentration of azithromycin was determined in accordance with CLSI guidelines using the agar dilution method on all isolates with halo diameter <15 mm. We have added additional text in the methods section (**Lines 907-924**) and clarified that the azithromycin resistance in *E. coli* used in this manuscript may not correspond to clinically relevant phenotypic resistance.
- **Line 420:** defining “temporary” and “persistent” colonization must be made in light of a recent report (Kantele et al, Lancet Microbe, 2021) showing with daily sampling in international travelers that culture-based ESBL-PE colonization status varied on a day-to-day basis.
Response: We appreciate the reviewer pointing us to this study, and have referenced its findings in our text (**Lines 413-417**)
- **Line 427:** I think instead of “highlighted”, should be “font colored”.
Response: We have changed the text “highlighted” to “font colored”
- **Line 435**—does the SNV stability decrease around diarrhea in TD? Perhaps the lack of stability is due to a more virulent strain establishing and/or replacing a more innocuous strain.
Response: We agree with the reviewer that SNV instability in TD subjects compared to HT subjects is likely an effect of diarrhea where commensal *E. coli* are replaced by Diarrheagenic *E.coli* (DEC). We observed a similar pattern in our microbiome analysis as well where gut microbial stability largely remain unaffected for HT and TD subjects except during diarrheal episodes when large-scale disruptions occurred, (as measured by “shift” events) (Extended data table 6). In the current study, we cannot explicitly confirm the same for the stability of DEC isolates as a high-resolution sample set around diarrheal episodes would be required to perform a similar analysis. We have revised the manuscript text to reflect this point (**Lines 406-412**).

- **Figure 1a legend:** to be consistent with other legends, should offer an explanation of the figure, instead of interpretation. Are the Shannon/Bray Curtis measurements from reads collapsed at a specific taxonomic level? (This applies to several other subfigures) Figure 1 or supplement should also show richness, before, during and after diarrhea
Response: We have updated legends and captions of all figures to include more explanatory information instead of interpretation.
- **Figure 1F,** most of the figure is not addressed in the text or legend.
Response: We have referenced figure 1F in the revised text.
- **Figure 2a legend,** should the adjectives, “left” and “right” really refer to “top” and “bottom”?
Response: We have corrected figure 2a.
- **Ext data figure 2e:** Authors should specifically state that the percentages correspond to percent resistant.
Response: As per the reviewer’s suggestion, we have updated the figure caption.
- **Figure 3 legend**—type “a)” appears twice. B and C are potentially mislabeled as well
Response: Thank you for catching this typo, we have corrected the figure and caption.
- **Figure 3e;** is this only in diarrhea samples? Text makes it appear that way, but figure legend doesn’t specify. Because proteobacteria levels return to normal after diarrhea, but AMR abundance stays elevated, this conflicts with the statement, “...indicating that the majority of AR determinants in diarrheal samples are likely concentrated within Enterobacteriaceae species.” To support the suggestion that AMR genes are concentrated in Enterobacteriaceae, it would be nice to see AMR abundance in HT with/without detected Enterobacteriaceae pathogen
Response: We clarify that not all ARGs are concentrated in Enterobacteriaceae, but that the correlation between increased ARG abundance in diarrheal samples with increased *Enterobacteriaceae* abundance—even as overall microbial diversity is decreasing—suggests that *Enterobacteriaceae* are carrying these ARGs.
- **Ext Figure 3d.** What statistical methods were used? There is a general lack of explanation in the figure legends regarding statistical testing.
Response: We have updated all the figures and captions with more explanation and included statistical tests that were used in the analysis.
- **Figure 5a legend** has a typo “shows”. Please provide a key for the different colors. Does the increased ARG in non-diarrheal TD vs HT samples hold if you look at samples before diarrhea (or is it driven by samples after diarrhea)?

Response: We have updated Figure 5a to include pre- and post- non diarrheal TD samples. We found that DEC stains isolated from diarrhea TD samples harbored more ARGs when compared to non-diarrhea TD and non-diarrhea HT samples.

We have updated the text and corrected the figure to include legends.

- **Ext Fig 7b and 7d**, are not mentioned in the text

Response: We have removed these figures as we did not discuss this analysis in the manuscript.

- **Ext Fig 9:** This figure is hard to understand—if they key showed SNVs associated with the heatmap it might be easier to follow. Parts of this figure aren't referenced in the text either. It seems like there are figures to characterize virulence factors that were not discussed in the main text.

Response: VFs were identified using the same pipeline as ARGs (line 1130). They, along with ARG heatmap, were included to support claims of clonality (i.e. isolates with same VF profile across timepoints) and simultaneous colonization (i.e. different VF profiles in same timepoint). The text and figure legends have been edited to reflect this.

Reviewer #2 (Remarks to the Author):

Boolchandani et al. describe a longitudinal study with 159 international students traveling to Peru. The study describes how travel changes the gut microbiota composition and how traveler's diarrhea impacts the species abundances in the gut. The authors have also investigated the antibiotic resistance gene profile shifts that occur alongside the changes observed in the microbiota composition. The study uses functional metagenomics, shotgun metagenomics, and genomic sequencing of strain and has produced a wealth of information to support the paper's main findings. The study complements the previous literature on the effect of travel on the antibiotic resistome, as previous studies have been limited by only including pre-and post-travel stool samples.

We thank the reviewer for their accurate summary of our work.

General comments:

The research has been done meticulously using appropriate methods and statistical approaches, and the sampling has been well planned. The analyses support the results and conclusions well. The article's language does not need additional revising.

We thank the reviewer for their kind comments.

The effect of travel on the resistome of travelers has been studied before. It is for example known that travelling to risk areas increases the odds of acquiring MDR pathogens. You could emphasize even more clearly how the results of the longitudinal sampling in a large cohort and the detailed analyses in your manuscript add to the existing literature. Highlighting the "most citable" items and results in your manuscript would improve the readability and make the paper more focused. Please review the manuscript critically and try to make the text more concise. Now there are a wealth of results reported that do not necessarily support the main message of the paper, which might be slightly lost from the readers. Some results could be moved to the supplement or left out.

We thank the reviewer for the suggestion. To improve readability and make the paper more focused, we have moved descriptions of the following isolate analyses to the "Supplementary Results" section: multiplex PCR, AST, WGS and phylogeny, and ARGs. Additionally, we have cut the section on isolate virulence factors.

Please include sample sizes, p-values, q-values, or FDR values when reporting results of statistical tests and the name of the test used in the results section. Also, include 95 % confidence intervals and R², or similar metrics whenever applicable.

We have now revised the results section of the manuscript to include complete statistical information.

More detailed comments:

- **Lines 203-209:** I am having a hard time following the train of thought in these sentences. Could the authors please revise and maybe shorten the sentences or simplify the language to make this section more easily understandable?

Response: We have reworded the paragraph to: *"Thus, we sought to determine how experiencing diarrhea affects this divergence from baseline, and whether increased divergence might be a predictor of who will get diarrhea. We compared the Bray-Curtis dissimilarities of HT and TD subjects' 1st week baseline sample with a non-diarrheal sample collected 1 month later. We further sub-divided the TD group between those who experienced diarrhea before 1 month (Early TD) or after 1 month (Late TD). Early TD subjects had significantly higher dissimilarity after 1 month of travel than the other groups (Fig. 1e). This suggests that while all subject's microbiomes continuously diverge from baseline during travel, diarrhea is an impactful perturbation which significantly increases this divergence. Further, we observed no significant difference between HT and Late TD subjects (Fig. 1e). This suggests that the divergence of Late TD subjects prior to diarrhea is indistinguishable from those who will not get diarrhea (HT), and that this metric cannot be used as an early predictor of who will get diarrhea."* (Lines 200-212)

- **Lines 217-218:** Please report the p-value or FDR cut-off used to identify species which were significantly differentially abundant.

Response: We have updated the figure and text to include FDR values

- **Line 243:** Could you clarify how SparCC or Netshift identifies which taxa are enriched between HT and TD subjects?

Response: We do not claim that these tools identified taxa enriched between HT and TD subjects, but between diarrhea and non-diarrhea sample types.

- **Lines 239-253:** Could you briefly clarify how "gained/lost connectivity" and "gained interactions" are measured? Is it tested statistically somehow, or are you comparing the number of connections of a species?

Response: As described in Methods, the diarrhea and non-diarrhea networks were constructed with SparCC using only significant connections (bootstrapped correlation p-values <0.05). Next, networks were compared using NetShift to identify taxa with a) an altered set of associated partners (high NESH score), and b) having greater connectivity to other nodes in the network (higher betweenness centrality score).

- **Lines 319- 391:** Could this section or a part of it be shortened or added to supplemental results? This section has parts that are not crucial for understanding the main message of the manuscript, and the manuscript could be made more compact by moving some of the text to the supplement.

Response: As per reviewer's suggestion, portions of these sections were moved to the Supplementary Results section.

- **Lines 419:** Why are you performing pairwise co-occurrence comparisons if you have genome assemblies? Isn't it possible to see from the contigs which ARGs co-occur? Or what does "pairwise co-occurrence comparison" mean in this context? Are you utilizing the assembled contigs? Please clarify.

Response: We have edited the text to clarify that in this context pairwise co-occurrence comparison indeed means ARGs found in the same contig (within 5 kb).

- **Lines 1212-1260:** Figure legend texts on these lines and below the figures themselves are different in at least their numbering. Please revise.

Response: Thank you for catching this error. We have revised the figure legends and the manuscript text to reflect the correct numbering.

- **Figure 1f:** Figure guide with $-\log(qval)*\text{sign}(\text{coef})$ needs spelling out of the abbreviations in the figure legend and some explanation in how to interpret the scale since this is a non-standard way of showing differential abundances. For example, what does -6 mean? Is it a q-value of 10^{-6} and a depleted species? Are all of these taxa shown significantly differentially abundance, and what is their q-value?

Response: The $(-\log_{10}(qval)*\text{sign}(\text{coef}))$ is the log normalized FDR value. This value is reported by MaAsLin2⁶ and it makes the q-value more interpretable and the sign shows the direction of the significant association. For example, if a feature is negatively associated

with a phenotype with FDR value of 1e-05, then the log normalized value of that feature would be -5 $[-\log_{10}(1e-5)*(-)]$. We have also updated the figure and its legend to include this information

- **Figure 1f and also elsewhere:** Please use either qvalue or FDR throughout the paper since they mean the same thing (or clarify how they differ from each other).

Response: We have corrected the discrepancy and now use FDR throughout the manuscript.

- **Figure 1g:** Why is there a p-value in the figure and not FDR or q-value? What test was used to calculate this.

Response: In Fig. 1f -1g we used MaAsLin2⁶, a multivariable statistical framework that performs generalized linear and mixed models to identify associations between microbial features and metadata variables of interest while controlling for covariates (*e.g.*, age, sex, region, and antibiotics) and inter-individual variability within subjects. Significant associations were then subjected to multiple hypothesis testing correction using the Benjamini-Hochberg method with an FDR threshold of 0.25.

- **Text referencing Figure 1f and 1g:** Please include the name of the test and p-value, q-value, or FDR value.

Response: We have revised all figures and captions to add more information about statistical tests used in the analysis.

- **Figure 2a:** It is challenging to compare the node sizes to each other. Please include a guide for the size of the node and use categories or transformations for sizes to make it easier to see. You could also omit the abundance information from the figure.

Response: We have revised Figure 2a to include legend for node size.

- **Figure 2a:** What are the right and left networks? Aren't the networks arranged top and bottom?

Response: We have corrected the labels in Figure 2a.

- **Figure 2b:** Please include a guide for the node sizes. Please spell out NESH score in the figure legend.

Response: As per the reviewer's suggestion, we have corrected the figure to include guide for the node sizes

- **Figure 2c:** Could you give an example of how to interpret the z-scores? If the z-score is -3, what does it mean?

Response: Figure 2c was generated by SIAMCAT⁷, a machine learning toolbox designed for comparative metagenomics. In our study, we used it to identify diarrhea-associated key taxa by comparing non-diarrheal PreTD and diarrheal TD samples. The heatmap in the figure displays the z-score normalized abundance of species across all samples, where z-

score > 0 means species is enriched and z-score < 0 means depleted in diarrhea TD compared to non-diarrheal PreTD samples.

- **Figure 3f:** What does log normalized FDR value mean in the scale? How should I interpret value -5? Are these all significantly different, and what is the test? There are two panels within the f-panel, but only one of them is discussed in the figure legend. Please include text for the other panel (left side) or remove this part of 3f.

Response: The log normalized FDR value shown in the heatmap is calculated by using the following formula: $(-\log_{10}(qval) * \text{sign}(\beta))$. This value is reported by MaAsLin2⁶ and it makes the q-value more interpretable and the $\text{sign}(\beta)$ shows the direction of the significant association. For example, if a feature is negatively associated with a phenotype with FDR value of $1e-05$, then the log normalized value of that feature would be -5 $[-\log_{10}(1e-5) * (-)]$. We have updated the figure and its legend to include this information.

- **Figure 3g:** Why are these p-values and not FDR values? Should these be corrected for multiple testing as there are several genes you were testing?

Response: We have replaced the p-value with FDR values in Fig 3g.

References

- 1 Feng, Z. *et al.* A human stool-derived *Bilophila wadsworthia* strain caused systemic inflammation in specific-pathogen-free mice. *Gut Pathog* **9**, 59, doi:10.1186/s13099-017-0208-7 (2017).
- 2 Hyams, J. S. *et al.* Clinical and biological predictors of response to standardised paediatric colitis therapy (PROTECT): a multicentre inception cohort study. *Lancet* **393**, 1708-1720, doi:10.1016/s0140-6736(18)32592-3 (2019).
- 3 Cheng, Y., Ling, Z. & Li, L. The Intestinal Microbiota and Colorectal Cancer. *Front Immunol* **11**, 615056, doi:10.3389/fimmu.2020.615056 (2020).
- 4 Jennings, M. C. *et al.* Case-Case Analysis Using 7 Years of Travelers' Diarrhea Surveillance Data: Preventive and Travel Medicine Applications in Cusco, Peru. *Am J Trop Med Hyg* **96**, 1097-1106, doi:10.4269/ajtmh.16-0633 (2017).
- 5 Ochoa, T. J. *et al.* High frequency of antimicrobial drug resistance of diarrheagenic *Escherichia coli* in infants in Peru. *Am J Trop Med Hyg* **81**, 296-301 (2009).
- 6 Mallick, H. *et al.* Multivariable association discovery in population-scale meta-omics studies. *PLoS computational biology* **17**, e1009442, doi:10.1371/journal.pcbi.1009442 (2021).
- 7 Wirbel, J. *et al.* Microbiome meta-analysis and cross-disease comparison enabled by the SIAMCAT machine learning toolbox. *Genome Biol* **22**, 93, doi:10.1186/s13059-021-02306-1 (2021).

REVIEWER COMMENTS

Reviewer #1 (Remarks to the Author):

The manuscript is much improved, and has addressed nearly all the prior critiques. Two major points remained incompletely addressed:

The authors addressed the first Major Point by using a linear mixed effect model to account for antibiotic usage, among other parameters. However, I would like to see the main analyses be stratified between TD with AbxUse vs. TD without ABxUse, or at least a sensitivity analysis looking at TD without ABxUse only. This is to support the statement in the abstract that, "diarrhea disrupted this stability and resulted in an increased abundance of antimicrobial resistance genes that can remain high for weeks." Otherwise, it is difficult to determine whether it was the ABxUse or the TD that disrupted the stability

The second Major Point, regarding antimalarial use, was not addressed appropriately (or at least I could not find this information in the supplementary notes). For example, perhaps the most common antimalarial prevention used is daily doxycycline, which have antibacterial effects. How many of the study subjects took daily doxycycline for malaria prevention during the study? I would like to see these excluded (or at least a sensitivity analysis taking these individuals out).

Minor:

Line 393 - last word is mis-spelled (timpeints)

Figure 3 is missing subfigure "b"

Reviewer #2 (Remarks to the Author):

The authors have made the corrections and modifications I suggested, and I am pleased to recommend the publication of the revised manuscript.

REVIEWER COMMENTS

Reviewer #1 (Remarks to the Author):

The manuscript is much improved and has addressed nearly all the prior critiques. Two major points remained incompletely addressed:

Major #1:

The authors addressed the first Major Point by using a linear mixed effect model to account for antibiotic usage, among other parameters. However, I would like to see the main analyses be stratified between TD with AbxUse vs. TD without ABxUse, or at least a sensitivity analysis looking at TD without ABxUse only. This is to support the statement in the abstract that, “**diarrhea disrupted this stability and resulted in an increased abundance of antimicrobial resistance genes that can remain high for weeks.**” Otherwise, it is difficult to determine whether it was the ABxUse or the TD that disrupted the stability

Response: As per reviewer’s recommendation, we re-performed the sensitivity analysis with only participants who didn’t take antibiotics at any point during their stay in Peru. Below we have shown that our main conclusions still hold true on the smaller cohort of individuals (N=111, HT=41, TD=70) (i.e., without antibiotics usage) (Figure R1) and the results are similar to our original analysis where we controlled for antibiotics-use and other parameters (age, gender, region, and subjects) using linear mixed effect models.

Figure R1 a) The gut microbial composition changed significantly throughout travel as measured by Bray-Curtis dissimilarities between each subjects' samples and their 1st-week baseline sample. **b)** The taxonomic diversity remains stable as measured by Shannon index over time. **c, d)** Diarrhea alters the gut microbial composition and resulted in increased abundance of antimicrobial resistance genes that remains high in post-diarrhea samples.

Major #2:

The second Major Point, regarding antimalarial use, was not addressed appropriately (or at least I could not find this information in the supplementary notes). For example, perhaps the most common antimalarial prevention used is daily doxycycline, which have antibacterial effects. How many of the study subjects took daily doxycycline for malaria prevention during the study? I would like to see these excluded (or at least a sensitivity analysis taking these individuals out).

Response: We apologize for not making it clear in the previous revision where we only focused on the diarrheal episodes. At the time of enrollment, participants were asked to report the intake of any medication that they had taken (either for illness or prophylaxis) within 2 weeks of enrollment. Among 159 participants, 7 individuals reported the intake of antibiotics for illness and 5 individuals took medication for prophylaxis (Doxycycline, n=1; Malarone, n=3; Ciprofloxacin, n=1). We have included this information in **Extended Data Table 1**. Also, as per reviewer's recommendation, **we excluded these individuals when we performed the sensitivity analysis on individuals with "No Antibiotics"** (see Major point #1; Figure R1) and showed that our main conclusions of the manuscript still hold true.

Minor #1:

Line 393 - last word is mis-spelled (timepoints)

Response: Thank you for catching this textual error. We have corrected the text in the revised manuscript.

Minor #2:

Figure 3 is missing subfigure “b”

Response: Thank you for catching this error. We have corrected the labels in the Figure 3

Reviewer #2 (Remarks to the Author):

The authors have made the corrections and modifications I suggested, and I am pleased to recommend the publication of the revised manuscript.

We thank the reviewer for recommending the manuscript.

REVIEWERS' COMMENTS

Reviewer #1 (Remarks to the Author):

I am satisfied with the revisions addressing the critiques.